

# Evaluation of Mei-yu Heavy-Rainfall Quantitative Precipitation Forecasts in Taiwan by A Cloud-Resolving Model for Three Seasons of 2012-2014

**Chung-Chieh Wang[1], Pi-Yu Chuang[1*], Chih-Sheng Chang[1], Kazuhisa Tsuboki[2],**
**Shin-Yi Huang[1], and Guo-Chen Leu[3]**
[1] Department of Earth Sciences, National Taiwan Normal University, Taipei, Taiwan
[2] Institute for Space-Earth Environmental Research, Nagoya University, Nagoya, Japan
[3] Central Weather Bureau, Taipei, Taiwan
Corresponding author: Pi-Yu Chuang (giselle780507@hotmail.com), Department of Earth
Sciences, National Taiwan Normal University. No. 88, Sec. 4, Ting-Chou Rd., Taipei, 11677
Taiwan.



**Abstract**
In this study, the performance of quantitative precipitation forecasts (QPFs) by the Cloud-
Resolving Storm Simulator (CReSS) in real-time in Taiwan, at a horizontal grid spacing of 2.5
km and a domain size of $1500 \times 1200$ km$^2$, within a range of 72 h during three mei-yu seasons of
2012-2014 is evaluated using categorical statistics, with an emphasis on heavy events ($\geq$ 100 mm
per 24 h). The overall threat scores (TSs) of QPFs for all events on day 1 (0-24 h) are 0.18, 0.15,
and 0.09 at the threshold of 100, 250, and 500 mm, respectively, and indicate considerable
improvements compared to past results and 5-km models.
Moreover, the TSs are shown to be higher and the model more skillful in predicting larger
events, in agreement with earlier findings for typhoons. After classification based on observed
rainfall, the TSs of day-1 QPFs for the largest 4% of events by CReSS at 100, 250, and 500 mm
(per 24 h) are 0.34, 0.24, and 0.16, respectively, and can reach 0.15 at 250 mm on day 2 (24-48
h) and 130 mm on day 3 (48-72 h). The larger events also exhibit higher probability of detection
and lower false alarm ratio than weaker events almost without exception across all thresholds.
The strength of the model lies mainly in the topographic rainfall in Taiwan rather than
migratory events that are less predictable. Our results highlight the crucial importance of cloud-
resolving capability and the size of fine mesh for heavy-rainfall QPFs in Taiwan.

## 1 Introduction

The quantitative precipitation forecast (QPF) is one of the most challenging areas in modern numerical weather prediction (NWP; e.g., Golding, 2000; Fritsch and Carbone, 2004; Cuo et al., 2011), especially for extreme events that have high potential for hazards. With its steep and complex topography, Taiwan over the western North Pacific (Fig. 1) experiences extreme rainfall rather frequently mainly in two periods: the typhoon (July-October) and mei-yu (May-June) seasons (e.g., Kuo and Chen, 1990; Wu and Kuo, 1999; Jou et al., 2011; Chang et al., 2013), where landslides and flash floods in/near the mountains and flooding over low-lying plains and urban areas are the main hazards (e.g., Wang et al., 2012b, 2013b, 2016b). In order to better prepare for these hazards and reduce their impacts, model QPFs and their verifications, especially over heavy-rainfall thresholds from large events, are thus very important for Taiwan. Of course, to identify where the model can make significant improvements in QPFs and what approaches are effective to achieve them are also crucial (e.g., Clark et al., 2011).

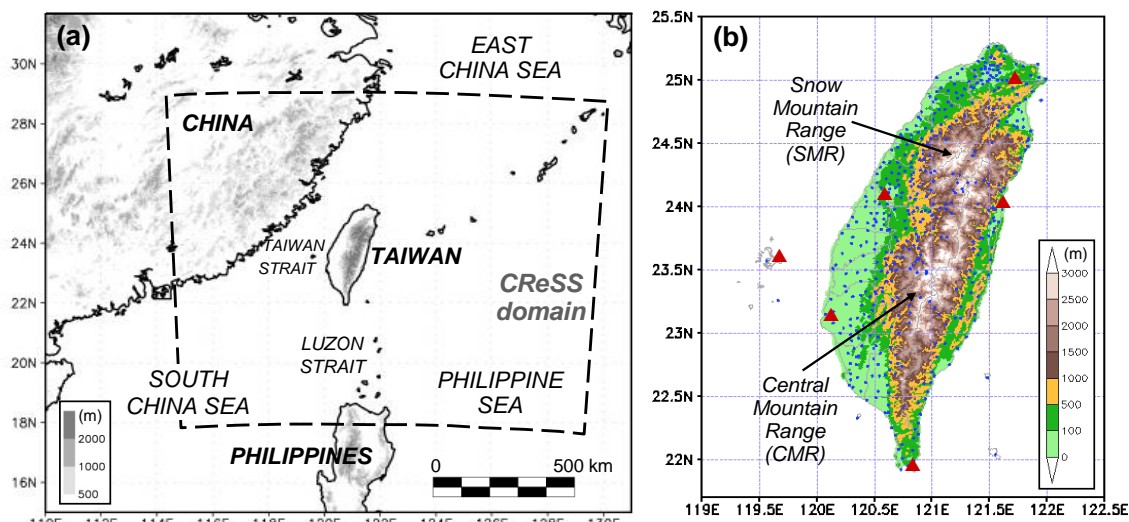

**Figure 1.** (a) The geography and topography (m, shading) surrounding Taiwan and the domain of 2.5-km CReSS (thick dashed box), and (b) the detailed terrain of Taiwan (m, color) and the locations of rain gauges (blue dots) and land-based radars (scarlet triangles) used to produce the reflectivity composites by the Central Weather Bureau (CWB). The two major mountain ranges in Taiwan, the Central Mountain Range (CMR) and Snow Mountain Range (SMR), are marked in (b).



For the mei-yu season in Taiwan, earlier studies mainly employed the widely-used, standard
categorical measures (see Section 2.4) to evaluate the performance of models such as the
Mesoscale Model version 5 (MM5) at thresholds up to 50 mm per 12 h (e.g., Chien et al., 2002,
2006; Yang et al., 2004). Their results show that the models at the time had some skill in
predicting rainfall occurrence at thresholds ≤ 2.5 mm, but little skill at 50 mm and above. In
recent years, several studies (e.g., Hsu et al., 2014; Li and Hong, 2014; Su et al., 2016; Huang et
al., 2016) have also examined the QPFs by the Weather Research and Forecasting (WRF) model
(Skamarock et al., 2005) running at the Central Weather Bureau (CWB), including its ensembles
at 5-km grid spacing ($\Delta x$). These studies indicate improvements over earlier models at thresholds
up to 50-100 mm (per 12 h) over the previous decade. However, the skill at 150-200 mm and
beyond is still limited, even with probability-matching (e.g., Ebert, 2001) within the forecast
range of 24 h (see e.g., Figs. 9 and 10 of Huang et al., 2016). While the scores at the CWB will
be compared with our results later, effective strategies and methods to improve the skill level of
NWPs at thresholds near 100-150 mm and beyond are obviously much needed.
Wang (2015, hereafter referred to as W15) evaluated the QPFs, within 3 days, by a cloud-
resolving model (CRM), the Cloud-Resolving Storm Simulator (CReSS; Tsuboki and
Sakakibara, 2002, 2007), for all 15 typhoons to hit Taiwan in 2010-2012. With $\Delta x = 2.5$ km, a
grid size comparable to research (e.g., Wang et al., 2005, 2011; 2013a, also Bryan et al., 2003;
Done et al., 2004; Clark et al., 2007; Roberts and Lean, 2007), these deterministic forecasts show
superior performance in QPFs, with threat scores (TSs, defined in Section 2.4) of 0.38, 0.32, and
0.16 at thresholds of 100, 250, and 500 mm, respectively, for all typhoons on day 1 (0-24 h, cf.
his Fig. 13). Even on day 3 (48-72 h), the corresponding TSs are 0.21, 0.12, and 0.01. Thus, the
skill by this CRM over the thresholds of 100-500 mm is remarkably high for typhoon rainfall in
Taiwan.
Moreover, as summarized in Wang (2016, hereafter W16), W15 found a strong positive
dependency of categorical scores on overall rainfall amount (or event magnitude). That is, the
more rain, the higher the scores, and the better the model performs. For example, the TSs at the
same thresholds (100, 250, and 500 mm) for his top-5 events (roughly top 5%) on day 1 are 0.68,
0.49, and 0.24, respectively (Fig. 1 of W16), all at least 1.5 times higher than their counterparts
for all typhoons. An important implication of this finding is that the model QPFs for extreme
events, *may not* be accurately assessed through categorical statistics without proper classification



to isolate them from ordinary events, and particularly not by taking arithmetic mean of TSs of
multiple forecasts. Wang (2015) also predicts the dependency, as a fundamental property, to
exist in other rainfall regimes. For mei-yu rainfall in Taiwan, we are certainly keen to find out
how this CRM performs, especially for the extreme events. Therefore, the main purpose of the
current study is three-fold: 1) to assess the skill of the 2.5-km CReSS in predicting mei-yu
rainfall, 2) to clarify whether the dependency property in categorical scores in the mei-yu regime
in Taiwan and further evaluate the model QPFs for larger and extreme events, and 3) if the QPFs
by CReSS prove to be superior to those reviewed above, why or where its strength lies? To
answer these questions above are our objectives.

In Section 2, the model, data, and methodology are described. In Section 3, the overall

scores of QPFs for groups with different event magnitudes are presented and compared with
previous results. Then in Section 4, examples are selected to illustrate how the CRM performs in
real-time forecasts and where its strength lies. Aspects related to the dependency property are
further discussed in Section 5, and our conclusions are given in Section 6.

**2 Data and Methodology**

2.1 The CReSS model and its forecasts

The CReSS model is a non-hydrostatic, compressible CRM with a single domain and no

nesting (Tsuboki and Sakakibara, 2002, 2007), and it has been used for weather forecasts in
Taiwan since 2006 (http://cressfcst.es.ntnu.edu.tw/, W15; Wang et al., 2013, 2016a). Starting
from July 2010, a grid size of 2.5 km is utilized, with a domain of $1500 \times 1200$ km$^2$ since May
2012 (Fig. 1a and Table 1). In CReSS, cloud formation, development, and all related processes
are explicitly treated using a bulk cold-rain microphysical scheme with six species: vapor, cloud
water, cloud ice, rain, snow, and graupel without any cumulus parameterization scheme. Other
sub-grid scale processes parameterized in the model include turbulent mixing in the planetary
boundary layer, as well as surface radiation and momentum/energy fluxes. These physical
options are identical to W15, and also given in Table 1.

The operational analyses and forecasts by the Global Forecasting System (GFS, Kanamitsu,

1989; Kalnay et al., 1990; Moorthi et al., 2001; Kleist et al., 2009) of the National Centers for
Environmental Prediction (NCEP), produced every 6 h (at 26 levels) were used as initial and


boundary conditions (IC/BCs) for CReSS (Table 1), which are also run four times a day, each
out to 72 h (now 78 h). At the lower boundary, terrain data at 30" resolution (roughly 900 m) and
the NCEP analyzed sea surface temperature (SST) are also provided. With its limited domain
size, the atmospheric evolution in CReSS is highly dictated by the NCEP forecasts, especially at
longer ranges. Note that since 2013, the IC/BCs from the GFS have doubled the resolution from
$1° \times 1°$ to $0.5° \times 0.5°$, but all other settings are kept the same during our study period (Table 1).

**Table 1.** The basic configuration, initial/boundary conditions (IC/BCs), and physical packages of the 2.5-km CReSS
used for real-time operation in 2012-2014.

| Season | 2012 | 2013 and 2014 |
|---|---|---|
| Projection | Lambert Conformal (center at 120°E, secant at 10°N and 40°N) | |
| Grid spacing (km) | $2.5 \times 2.5 \times 0.2\text{-}0.663$ (0.5)* | |
| Grid dimension ($x, y, z$) | $600 \times 480 \times 40$ | |
| Domain size (km) | $1500 \times 1200 \times 20$ | |
| Forecast frequency | Every 6 h (at 0000, 0600, 1200, and 1800 UTC) | |
| Forecast range | 72 h | 78 h |
| IC/BCs (including SST) | NCEP GFS analyses/forecasts (at 26 levels) | |
| | $1° \times 1°$ | $0.5° \times 0.5°$ |
| Topography | Real at $(1/120)°$ spatial resolution | |
| Cloud microphysics | Bulk cold-rain scheme (Lin et al., 1983; Cotton et al., 1986; Murakami, 1990; Ikawa and Saito, 1991; Murakami et al., 1994) | |
| PBL/turbulence | 1.5-order closure with prediction of turbulent kinetic energy (Deardorff, 1980; Tsuboki and Sakakibara, 2007) | |
| Surface processes | Energy/momentum fluxes, shortwave and longwave radiation (Kondo, 1976; Louis et al., 1982; Segami et al., 1989) | |
| Substrate model | 41 levels, every 5 cm to 2 m | |

* The vertical grid spacing of CReSS is stretched (smallest at bottom), and the averaged spacing is given in the
parentheses.

2.2 Data

The observational data used include synoptic weather maps from the CWB, the vertical

maximum indicator (VMI) reflectivity composites every 30 min from land-based radars, and
hourly rainfall data from about 440 gauges in Taiwan (Fig. 1b) for QPF verification. Along with
NCEP gridded final analyses (on $1° \times 1°$ grid), the weather maps are used to identify and



synthesize the occurrence of favorable factors to heavy rainfall among events with different
magnitude (to be described in Section 2.3). For selected heavy-rainfall cases, the radar
composites are compared with the CReSS forecasts to assess the quality of the QPFs in Section

4.

2.3 Verification period classification

In this study, objective categorical statistics (e.g., Schaefer, 1990; Wilks, 2011) are used to

verify QPFs mainly because: 1) the ability of models to predict the heavy rainfall at the correct
location is imperative in Taiwan, since its primary hazards are landslides and floods, and 2) our
results can be easily compared with earlier studies. Here, 24-h QPFs are chosen because: 1) the
bulk rainfall accumulation from mei-yu events, as for typhoons, is our main concern rather than
the rain over shorter periods, especially at longer ranges (days 2-3), and 2) the issue of double
penalty on high-resolution QPFs (e.g., Ebert and McBride, 2000) is less serious using a longer
accumulation period. Although the CReSS forecasts are made four times a day, only those from
0000 and 1200 UTC are evaluated in this study.

**Table 2.** The classification criteria using (at least) 10% of rain gauges with highest 24-h accumulated rainfall (0000-
2400 or 1200-1200 UTC) over Taiwan, and the results in the number of 24-h segments (and percentage) and total
points (sites) of $H + M + FA$ at selected rainfall thresholds (mm) for the different groups. During the mei-yu seasons
of 2012-2014, the total $N$ is 148776 and on average there are 442 rain gauges per segment. The points of $H + M +$
$FA$ are based on the statistics of day-1 (0-24 h) QPFs, and $N$ is also given (with no threshold).

| Group | Criterion (of 10% gauges) | No. of segments (%) | No. of all points ($N$) | No. of points ($H + M + FA$) at threshold (mm) 50 | 100 | 250 | 500 |
|---|---|---|---|---|---|---|---|
| A+ | ≥ 130 mm (a subset of A) | 13 (3.9) | 5622 | 3807 | 2453 | 490 | 32 |
| A | ≥ 50 mm | 61 (18.1) | 26826 | 11000 | 4889 | 675 | 47 |
| B | ≥ 25 mm, but not A | 75 (22.3) | 33018 | 4279 | 1078 | 98 | 4 |
| C | ≥ 10 mm, but not B | 88 (26.1) | 38583 | 1675 | 281 | 10 | 3 |
| D | ≥ 1 mm, but not C | 67 (19.9) | 29267 | 266 | 32 | 0 | 0 |
| X | < 1 mm | 46 (13.6) | 20067 | 59 | 20 | 4 | 0 |
| All | A through D plus X | 337 (100.0) | 147761 | 17279 | 6300 | 787 | 54 |




A total of 366 target segments (0000-2400 and 1200-1200 UTC) in May-June, 2012-2014
are classified into several groups based on the observed rainfall using the following criteria, as
summarized in Table 2. Groups A, B, C, and D are those periods with at least 10% rain gauges
reaching 50, 25-50, 10-25, and 1-10 mm, respectively, while group X is the remaining periods
with little or no rain. The full classification results (Table 3) give a total of 337 segments,
excluding those under typhoon influence. Groups A-D individually account for about 18-26%
and are comparable in sample size, while the driest group X is about 14% (Tables 2 and 3).
These five groups are exclusive to each other, and the results without classification will be
referred to as the "all" group. From group A, a subset of A+ that has ≥ 10% sites reaching 130
mm is identified and represents the most-rainy 4% in our sample with the highest hazard
potential. The spatial distribution of mean mei-yu rainfall per season in 2012-2014, with a peak
amount of about 1700 mm, is shown in Fig. 2 and resembles the climatology (e.g., Yeh and
Chen, 1998; Chien and Jou, 2004; Chi, 2006; Wang et al., 2017).

**Table 3.** The full classification result for all the 24-h verification periods during the three mei-yu seasons of 2012-
2014. For each month, the first (second) row gives the results of 0000-2400 (1200-1200) UTC. While the groups of
A-D and X are denoted by their corresponding letter, a bold A indicates group A+ (a subset of A) and T marks the
periods influenced by tropical cyclones and thus excluded from study.

| Year | Month | Time (UTC) | Date in month | | | Segments included (A-D, X) |
|------|-------|-----------|------|------|------|------|
| | | | 1-10 | 11-20 | 21-31 (or 21-30) | |
| 2012 | May | 0000 | XAABDXXXBB | CCXXCBAAAB | XXDXDCAAABC | 31 |
| | | 1200 | CAADXXXBCC | CXXCBBAB**A**D | XXXXDBAABBD | 31 |
| | Jun | 0000 | CCBCCCCB**AA** | **AAAAA**BBTTT | TCCDDCTTTD | 23 |
| | | 1200 | DBCDCDBA**AA** | **A**AAABBTTTT | TCDDDTTTTD | 21 |
| 2013 | May | 0000 | CCCCBBCCDA | AACCCABAA**A** | CBCCDDDDDX | 31 |
| | | 1200 | CCDCACCDD**A** | BBCCCABABA | BBCCDDDDDXX | 31 |
| | Jun | 0000 | XXCBCCXXCB | BBABCDDDXX | BDBCCXXXXD | 30 |
| | | 1200 | XDBCCDXDCB | BABBDDDXXB | CBBCXXXXXD | 30 |
| 2014 | May | 0000 | CCBBBBCCCD | DCBD**A**BXBA**A** | ADDCDCCBBBC | 31 |
| | | 1200 | CBDABCDCDD | CBDAAXBAA**A** | BDCCDBBABCD | 31 |
| | Jun | 0000 | XDACAABBBC | CTTTTTTDDB | DCBCCDXCAC | 24 |
| | | 1200 | XACBAABBDC | TTTTTTTDBD | CCBCDXDBAC | 23 |
| Total | | | A+: 13, A: 61, B: 75, C: 88, D: 67, X: 46 (T: 29) | | | 337 |


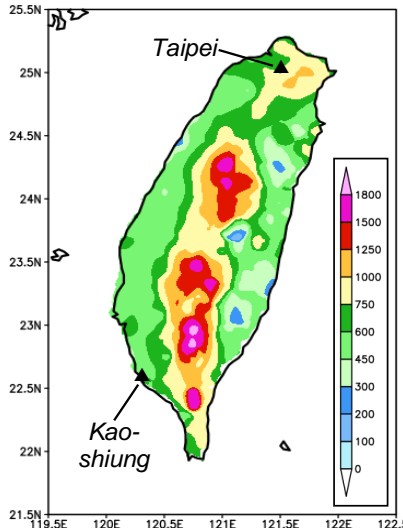

**Figure 2.** Spatial distribution of mean total rainfall (mm) per mei-yu season (1 May through 30 Jun) in 2012-2014.
The cities of Taipei and Kaoshiung are marked.

2.4 Categorical measures of model QPFs

As mentioned, the 24-h QPFs by CReSS are verified against the rain gauge data, at three

different ranges of 0-24, 24-48, and 48-72 h (days 1-3). For this purpose, objective skill scores
computed from the standard 2 × 2 contingency table (or the categorical matrix) at 14 thresholds
from 0.05 to 750 mm are adopted. These measures include the TS (also called critical success
index), bias score (BS), probability of detection (POD), and false alarm ratio (FAR), respectively
defined as (e.g., Schaefer, 1990; Wilks, 2011; Ebert et al., 2003; Barnes et al., 2009)

$TS = H/(H + M + FA),$                                            (1)

$BS = (H + FA)/(H + M) = F/O,$                              (2)

$POD = H/(H + M) = H/O,$ and                              (3)

$FAR = FA/(H + FA) = FA/F,$                              (4)

where $H$, $M$, and $FA$ are the counts of hits (both observed and predicted), misses (observed but
not predicted), and false alarms (predicted but not observed), respectively, among a total number
of $N$ verification points. Here, $N = H + M + FA + CN$, where $CN$ is the correct negatives (neither
observed nor predicted), and the total counts in observation ($O$) and forecast ($F$) are simply $O =$
$H + M$ and $F = H + FA$. The values of TS, POD, and FAR are all bounded by 0 and 1, and the



higher (lower) the better for TS and POD (FAR). For BS, its value can vary from 0 to ∞ (or $N -$
1 in practice), but unity is the most ideal. By interpolating the model QPFs onto the gauge sites
that serve as verification points (i.e., $N \approx 440$ per segment) using the bi-linear method, the counts
of $H$, $M$, $FA$, and $CN$ at any given threshold can be easily obtained for each segment. Although
the density of rain gauges varies to some extent (roughly every 5-10 km in the plains and ≥10-20
km in the mountains, cf. Fig. 1b), their weights are assumed equal (e.g., Wang, 2014). For any
group (e.g., A+) at a given threshold, the scores are obtained from a single $2 \times 2$ table that
combines the entries from all segments, so that the sample sizes are maximized (cf. Table 2, e.g.,
W15, W16). This practice also remedies the issue of sampling in-homogeneity and increases the
stability of results, especially toward the high thresholds, as long as the points involved in the
matrix are not too few in number (cf. Table 2). Since neither the observation nor the forecast
ever reached 750 mm (per 24 h) during the study period, results up to 500 mm (the next highest
threshold) are presented in later sections. Also, only 24-h QPFs are evaluated in the current
study. Except for the categorical matrix, subjective visual verification is also used in the selected
examples (Section 4).

**204 3 Mei-yu QPFs in 2012-2014**

3.1 Overall skill by the 2.5-km CReSS
Following the method described above, the categorical matrices across the thresholds are
obtained and the overall skill of CReSS in mei-yu QPFs during 2012-2014 is shown in Fig. 3
using the performance diagram. Proposed by Roebber (2009), the diagram uses the success ratio
(SR = 1 − FAR = $H/F$) and POD as its two axes, and can also depict the TS (gray curved
isopleths, higher toward upper-right) and BS (brown dotted lines) simultaneously. In Fig. 3, the
scores from forecasts at both 0000 and 1200 UTC for segments (of 24 h) in groups A+, A to D,
and all periods (A-D plus X, cf. Table 2) are shown for ranges of day 1, 2, and 3, respectively.
The "all" group (black) shows the overall skill for all mei-yu rainfall without classification, and
its TS for day-1 QPFs decreases slowly from 0.6 at 0.05 mm to 0.18 at 100 mm, 0.15 at 250 mm,
and 0.09 at 500 mm (Fig. 3a). Over heavy-rainfall thresholds ≥ 160 mm, the TSs of 0.09-0.16 are
significantly higher than those reviewed in section 1. Even on day 2, the TSs remain at 0.11 to
0.06 over 160-500 mm, and above 0.03 up to 350 mm on day 3 (Figs. 3c,e).



When all segments are stratified by the observed event magnitude, the TSs are higher and
the skill better for larger events than smaller ones, following the order of A+ then A to D for all
thresholds at all three ranges without any exception (Fig. 3), while each individual curve mostly
decreases with threshold when rain areas reduce in size (as shown in Fig. 4). Thus, the positive
dependency of categorical measures on rainfall amount is also strong and evident in mei-yu
QPFs in Taiwan, as predicted by W15. Linked to this dependency, the TSs for large events are
also higher than those for the "all" group from the entire sample. For the most hazardous group
A+, for example, the TS on day 1 is 0.34 at 100 mm, 0.24 at 250 mm, and 0.16 at 500 mm (per
24 h, Fig. 3a). On days 2 and 3, the corresponding TSs are 0.32, 0.15, and 0.07 (Fig. 3c), and
0.25, 0.05, and 0.00 (Fig. 3e), respectively, all higher than their counterparts for the all group
(except day 3 at 500 mm). If we select TS ≥ 0.15 to indicate some predictive skill, then the QPFs
by the 2.5-km CReSS have skill all the way up to 500 mm (per 24 h) on day 1, 250 mm on day 2,
and 130 mm on day 3. Also, for A+, A, and all groups, the TSs of day-2 QPFs stay quite close to
the values on day 1, and some are even identical, from low thresholds up to 200-250 mm. For
day-3 QPFs compared to day 2, the same is true up to about 130 mm (Fig. 3, left column). Such
results that some skill of heavy-rainfall QPFs still exists on days 2-3 are very encouraging. On
the other hand, at thresholds ≥ 50 mm, the skill for B-D events (Fig. 3, right column) are limited
(TS ≤ 0.08) when the rain areas are relatively small (with $O/N ≤ 6\%$, Fig. 4), but as discussed in
W15, this is not important due to low hazard potential.
Another fairly subtle but important feature in Fig. 3 is that the TSs of all, A, and A+ groups
decrease only marginally at times, or do not drop at all, across some heavy-rainfall thresholds,
particularly on days 1-2, despite the reduction in rain-area size (left column). Some examples
include the TSs for group A+ over 100-350 mm on day 1 (drops from 0.34 to 0.21), and those for
group A over the same thresholds on day 1 (from 0.23 to 0.15) and over 100-250 mm on day 2
(from 0.20 to 0.14). Even on day 3, the decrease of A and "all" curves from 160 to 350 mm is
rather slow, although the TSs there are only 0.03-0.07 (Fig. 3e). Such a slow decline in TSs with
thresholds indicates that in a relative sense, the model is more capable to produce hits toward the
rainfall maxima, which occur more frequently in the mountains (cf. Figs. 1b and 2).




**Figure 3.** Performance diagrams of 24-h QPFs for (a),(b) day 1 (0-24 h), (c),(d) day 2 (24-48 h), and (e),(f) day 3 (48-72 h) by the 2.5-km CReSS, at 13 rainfall thresholds (inserts) from 0.05 to 500 mm, during three mei-yu seasons (May-Jun) in 2012-2014 in Taiwan. Results for groups A+, A, and All (All, B, C, and D) are plotted in left (right) column with different colors. TS values (rounded to two decimal places) are labelled at fixed thresholds of 0.05, 50, 160, and 500 mm (open symbols) or selected endpoints (smaller fonts), and data points with TS = 0 at high thresholds are omitted. For each group, the threshold where the observed rain-area size ($O/N$) falls below 1% is labeled in insert, and also marked by an arrow in (a),(b).

By definition, both POD and SR cannot be lower than the TS [cf. Eqs. (1), (3), and (4)], and the ratio of POD/SR equals to the BS (thus, POD < SR if BS < 1 and vice versa). In Fig. 3, the PODs start at 0.05 mm from nearly perfect values of 0.98-0.99 for days 1-3 for group A+, at least 0.9 for A, and ≥ 0.72 for all segments (left column). For these three groups, the PODs at 250 mm remain at least 0.32 on day 1, 0.21 on day 2, and 0.05 on day 3. Like the TS, the skill in POD for mei-yu rainfall indeed decreases quite significantly with forecast range (lead time), particularly toward high thresholds, mainly due to error growth and the reduction in predictability, but some skill still exists at 130 mm even on day 3, with POD = 0.16 and TS = 0.07 (for all segments). The SR values (and thus FAR) are again the best for group A+ and ≥ 0.36 across all thresholds on day 1, including 500 mm (Fig. 3a). On day 2, the SRs for A+ over 130-500 mm decrease but not by too much, and the values over 10-250 mm even increase on day 3 (Figs. 3c,e). Often, the SR for A+ is considerably higher than those for A and all events regardless of forecast range, particularly over heavy-rainfall thresholds. Overall, the model also produces higher POD and SR (i.e., lower FAR) for larger events compared to smaller ones at all thresholds and all three forecast ranges in Fig. 3, with only a few exceptions after close inspection. Therefore, as for typhoon rainfall (W15), the 2.5-km CReSS is the most skillful in predicting the largest events in the mei-yu season in Taiwan..

Next, the BS values are examined for over/under-prediction (i.e., above/below the diagonal line) in Fig. 3, where the threshold with $O/N$ falling below 1% is marked to indicate values that might be potentially unstable and less meaningful. For day-1 QPFs, the BSs for all segments suggest slight under-prediction over low thresholds ≤ 10 mm (per 24 h), but some over-prediction (BS ≈ 1.25-1.5) across 50-350 mm (Fig. 3a). On the contrary, the model shows slight over-prediction over 0.05-75 mm for the largest events of group A+ (with BSs up to 1.15), but under-prediction toward higher thresholds, with the lowest BS of 0.52 at 350 mm. Mostly




between the two curves mentioned above, the curve for group A stays closer to unity and is more
ideal across nearly all thresholds (Fig. 3a). For B-D groups (Fig. 3b), their characteristics are
similar to the All group, with BSs of 0.8-1.0 at low thresholds but generally some over-
prediction across higher thresholds. However, their BS values rarely exceed 2.5, unless the $O/N$
values drop to below 1%. The situation for BSs between different groups remains similar on days
2 and 3 (Figs. 3c-f), and the over-forecasting across the middle thresholds in group A (at all
ranges) can be confirmed to come mainly from groups B-D, as groups A+ and A exhibit little or
a much less tendency for over-prediction there (Fig. 3).

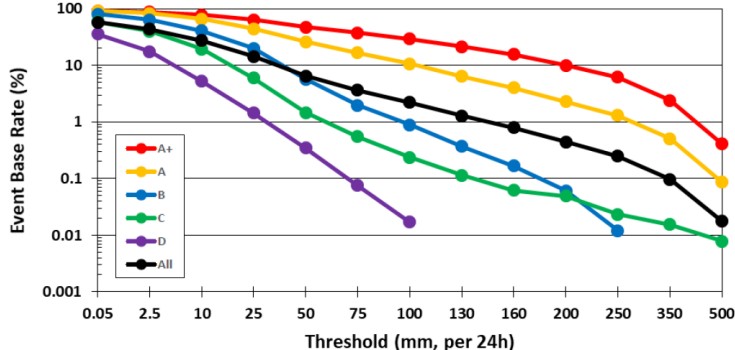

**Figure 4.** Observed rain-area size or base rate ($O/N$, %) of 24-h rainfall (same for days 1-3) in logarithmic scale
used to compute the scores in Fig. 3

Toward the longer ranges of days 2 and 3, the BS values in general become smaller,

particularly for the larger groups (Fig. 3, left column). Thus, the over-prediction in group A is
reduced and the under-prediction in A+, which is the most important group, becomes more
serious, especially toward the high thresholds (Figs. 3c,e). For example, the BS of day-2 QPFs
for A+ is ideal and ≥ 0.8 up to 200 mm but declines to about 0.35 at 500 mm, but it is already
below 0.4 at 130 mm on day 3. After closer inspection, the reason behind this behavior can be
revealed as the following. Since group A+ is the observed top events that turned out to be very
rainy, the atmosphere evolved in such a way to typically bring a greater number of favorable
factors together in synergy to result in their occurrence. In model forecasts, when errors grow
and the evolution deviates, the chance to become less rainy (not as favorable) is higher than more
rainy, more so in forecasts made earlier at longer ranges. Thus, the probability to under-forecast
peak rainfall rises with lead time. For smaller events that turned out to produce not much rainfall




(i.e., B-D and X), a similar tendency does not exist or is weaker, and BS tends to be greater than
unity. Thus, as exemplified here, the BSs from the larger and hazardous events can be quite
different from those from all events, which are inevitably affected by the more frequent but small
(unimportant) events. So, to say the least, one needs to practice caution in the interpretation of
BS, which can also become unstable when $O/N$ approaches zero (which inevitably happens at
certain thresholds).

3.2 Improvement in heavy-rainfall QPFs

To assess the improvement in heavy-rainfall QPFs in mei-yu season made by the 2.5-km

CReSS, our results are compared to those at the CWB for the same period. Figure 5 shows the
TSs of day-1 (0-24 h) and day-2 (24-48 h) QPFs for May and June of 2013 and 2014 by the
CWB WRF model and several products from their 20-member WRF ensemble prediction system
(WEPS, e.g., Hong et al., 2015), all with $\Delta x = 5$ km, as an example. These plots are produced for
each month at the CWB for routine verification (within a range of 48 h) since 2013, and are the
same as those used by Huang et al. (2015, 2016). In addition to deterministic forecasts, the scores
also include those using probability-matching techniques (PM and NPM, e.g., Ebert, 2001; Fang
and Kuo, 2013), which may provide some benefit over the ensemble mean (WEPS) over
thresholds of about 50-200 mm (e.g., Su et al., 2016; Huang et al., 2016). In Fig. 5, one can see
that the TSs are no higher than 0.07 at 100 mm (per 24 h) and 0.03 at 200 mm (and TS = 0 ≥ 300
mm) for either day 1 or day 2 in the two mei-yu seasons of 2013 and 2014, in line with the
review given in Section 1. Overall, the "all" curves in Fig. 3 indicate that the 2.5-km CReSS
exhibits better skill than those reviewed in Section 1 (with $\Delta x$ as fine as 5 km at most), especially
at thresholds above 100 mm (e.g., TS = 0.15 at 250 mm and 0.09 at 500 mm for day 1). With
even higher TSs for larger and more hazardous events (groups A and A+), the improvement of
heavy-rainfall QPFs in the present study from earlier results is therefore significant and quite
dramatic.






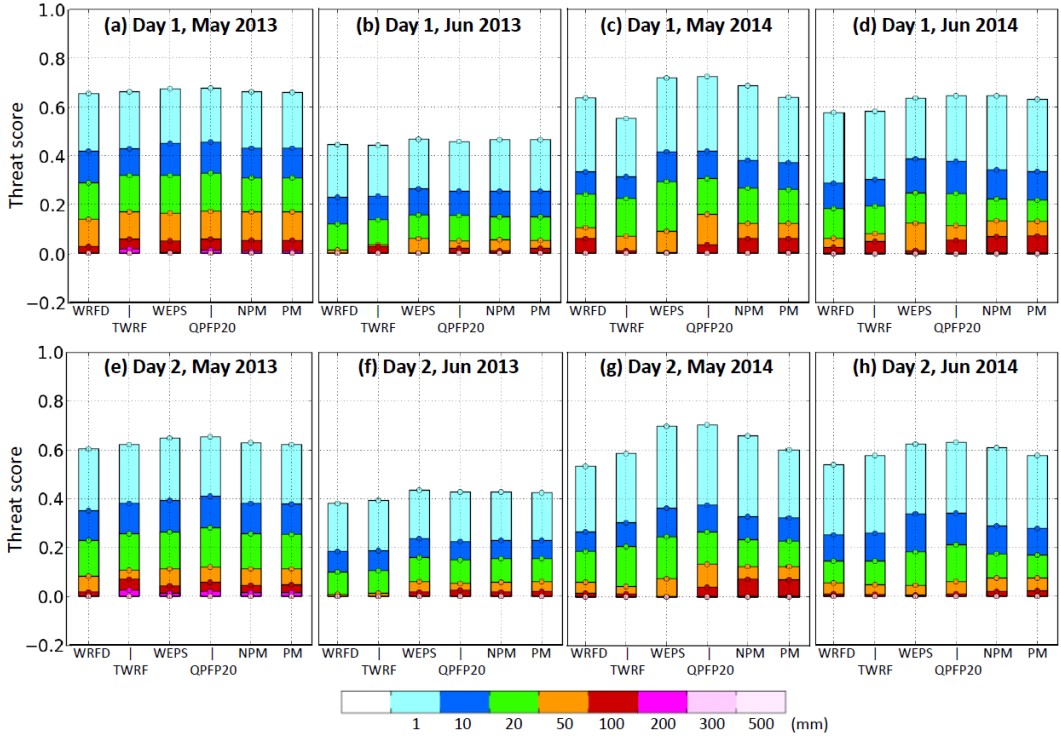

**Figure 5.** The TS of 0-24-h QPFs (day 1) for (a) May and (b) Jun of 2013, and (c) May and (d) Jun of 2014, respectively, at selected thresholds over 1-500 mm (per 24 h, scale at bottom) by two deterministic forecasts from WRF (WRFD) and the Typhoon WRF (TWRF) and four ensemble forecasts from the 20-member WRF Ensemble Prediction System: ensemble mean (WEPS), top 20% (QPFP20), and WEPS employing the probability matching (PM) and new PM (NPM) techniques. (e)-(h) As in (a)-(d), but showing the TS of 24-48-h QPFs (day 2), respectively.

## 4 Examples of Model QPFs

Given the success of the CRM in its overall performance shown above, some examples of CReSS forecasts are selected and presented in this section for further examination and discussion. The main goal here is two-fold: 1) To illustrate how the model behaves and captures the rainfall with the corresponding scores in individual forecasts in detail, and thus 2) to identify where such a CRM has high skill in QPF and where it has limitations in Taiwan, thereby to shed




light on the source of skill seen in Fig. 3. Since our focus is on heavy rainfall, the event during 9-
12 June 2012, the largest during our study period, is chosen for illustration.

The event of 9-12 June 2012 spanned four days and contributes more than half the segments

in group A+ (7 in 13, cf. Table 3), and the model's performance in predicting this event is thus
highly relevant. In Fig. 6a, the observed 24-h rainfall distributions over Taiwan are shown every
12 h, from 1200-1200 UTC 8 June to 0000-2400 UTC 12 June 2012. Except for the first forecast
period, all seven segments are qualified as A+ and five have a 24-h peak rainfall over 500 mm
(those from 1200 UTC 9 June). Reminiscent to the season average (cf. Fig. 2), three rainfall
maxima from this lengthy event exist: over southern CMR, near the intersection of CMR and
SMR in central Taiwan, and over northern Taiwan (Fig. 6a). The rain at the two mountain
centers (cf. Fig. 1b) is much more persistent than that in northern Taiwan, which concentrated
mainly over a 10-h period beginning 1400 UTC 11 June (Wang et al., 2016b). The southwesterly
plains also received considerable rainfall, especially around 9 June (Fig. 6a).

The 24-h QPFs produced by the 2.5-km CReSS (at 0000 or 1200 UTC) in real time

targeting the same periods as in Fig. 6a, at the ranges of days 1-3 are presented in Figs. 6b-d,
with the general quality expressed by the TS at 100 mm (lower right corner inside panels) and
thickened outline for TS ≥ 0.15 at the threshold of 50, 100, 200, 350, or 500 mm. The day-1
QPFs (Fig. 6b) are made from the forecasts starting (with initial time $t_0$) at the time of the
heading, while day-2 (Fig. 6c) and day-3 QPFs (Fig. 6d) in the same column (i.e., for the same
target period) are those made 24 and 48 h earlier, respectively. In Fig. 6, this extreme and
lengthy event was generally well captured by the model, especially on day 1 where the overall
rainfall pattern and TS both tend to be better, as expected. The best day-1 QPF is for 0000-2400
UTC 10 June (TS = 0.68 at 100 mm and 0.40 at 500 mm), followed by the one for 1200 UTC 11-
12 June (TS = 0.59 at 100 mm and 0.29 at 500 mm, columns 4 and 7, Fig. 6b). At longer ranges
on days 2 and 3, the rainfall magnitudes produced over the mountains and southwestern plains
are also comparable to observations, but the event starts somewhat too early and becomes less
rainy during 10-11 June with apparent under-forecast (Figs. 6c,d). As a result, the TSs for the
segments starting at 1200 UTC 8 June and during 10-11 June (columns 1 and 4-7) mostly
increase from longer to shorter ranges, i.e., with better QPFs at later times. This relationship with
range, however, does not hold true for the other segments, among which the day-3 and day-2



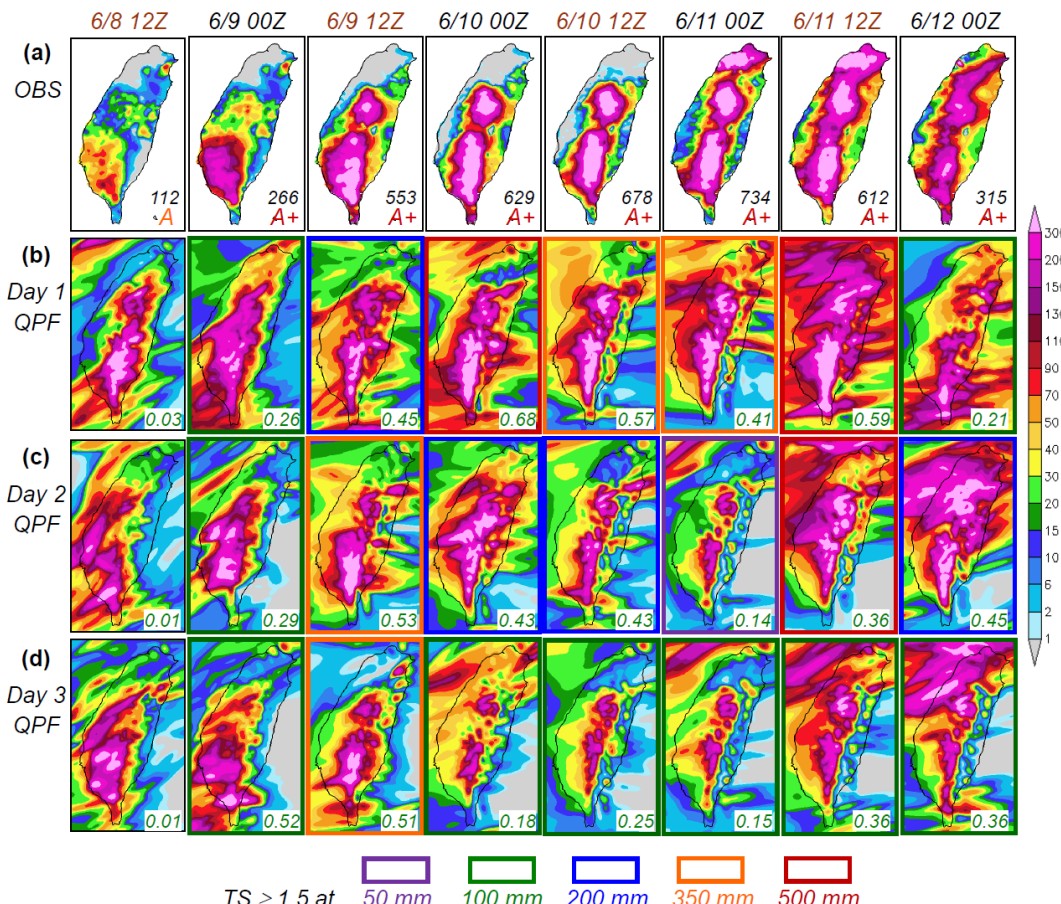

TS ≥ 1.5 at   50 mm   100 mm   200 mm   350 mm   500 mm

**Figure 6.** (a) The observed 24-h accumulated rainfall (mm, scale on the right) over Taiwan from 1200 UTC 8 Jun to

0000 UTC 13 Jun 2012, given every 12 h (from left to right), with the beginning time of accumulation (UTC)

labeled on top (black for 0000-2400 UTC and brown for 1200-1200 UTC). (b) Day-1 (0-24 h), (c) day-2 (24-48 h),

and (d) day-3 (48-72 h) QPFs valid for the same 24-h periods as shown in (a) by the 2.5-km CReSS (starting at

0000/1200 UTC under black/brown headings). In (a), peak 24-h rainfall (mm) and classification group are labeled.

In (b)-(d), thick boxes in purple, green, blue, orange, and scarlet denote forecasts having a TS ≥ 0.15 at the threshold

of 50, 100, 200, 350, and 500 mm (per 24 h), respectively, and the TS at 100 mm is also given (lower right corner).

QPFs for the period of 1200 UTC 9-10 June (TS ≥ 0.51-0.53 at 100 mm and 0.20-0.31 at 350

mm) and the day-2 QPF for 1200 UTC 11-12 June (TS = 0.40 at 500 mm) are particularly

impressive (columns 3 and 7). Compare to the rain over the terrain, the maximum across Taipei

in northern Taiwan during 11-12 June was largely over lower and flatter regions (cf. Figs. 1b and



2) and more challenging for the model to predict at the right location (Fig. 6), an aspect that will
be further elaborated on later.

Figure 7 shows the TS and BS of day-1 to day-3 QPFs from the runs made at a series of

initial times, including 1200 UTC of 7-9 June and the next four from 0000 UTC 10 to 1200 UTC
11 June (top to bottom), and our focus is mainly over the thresholds ≥ 100 mm. Inside the panels,
the observed event base rate ($O/N$, i.e., rain-area size, identical for the same target period) and
the hit probability ($H/N$, note that $H/N \leq O/N$) are given at selected points. With such
information, it is easy to work out forecast base rate ($F/N$), POD, and SR, and thus how the
model actually performed, particularly over the high thresholds. Some of the TSs at high
thresholds mentioned above in relation to Fig. 6 can also be verified here.

Figures 7a-7f provide some examples on how the model did in predicting the

commencement of the event (cf. Fig. 6, columns 1-3). As mentioned, the day-3 QPF made from
1200 UTC 7 June (Figs. 7a,b, blue curves) and day-2 QPF made one day later (Figs. 7c,d, red
curves), both targeting 1200 UTC 9-10 June, are of fairly high quality. With rain areas ($O/N$)
occupying 31%, 14%, and just 2% of Taiwan at 100, 200, and 350 mm, the day-2 QPF in Fig. 7c,
with BS ≈ 0.6-1.1 (Fig. 7d), yields TSs of 0.53-0.31 at these thresholds. The day-3 QPF with $t_0$ at
1200 UTC 7 June, with less predicted rain and BS ≈ 0.3-0.6 (cf. Fig. 6d, column 3), the TSs are
0.51-0.2 (Figs. 7a,b). With TS at least 0.2 at 350 mm (an amount predicted only in southern
CMR), both QPFs (for 1200 UTC 9-10 June) are quite skillful. Valid for periods with varying
magnitude (B, A, and A+), the forecasts in Figs. 7a,b are also good examples to illustrate the
dependency property (Fig. 3 and W15). In Figs. 7e,f, the TS curves at the three ranges (all for A+
events) are closer. In Fig. 8, the actual forecast near Taiwan between 42 and 69 h, from the run
made at 1200 UTC 7 June, is compared with radar observations every 6 h to examine general
rainfall locations. While a wind-shift line existed off eastern Taiwan, the surface mei-yu front
was well to the north with prefrontal low-level southwesterly flow impinging on the island
during this period (also Wang et al. 2016b). Active convection constantly developed over the
mountains in central and southern Taiwan and moved from the upstream ocean into the
southwestern plains, and this scenario was well captured by the 2.5-km CReSS (Fig. 8), yielding
a high-quality QPF on day 3 despite some under-forecast at thresholds ≥ 75 mm (cf. Figs. 7a,b).

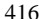

**Figure 7.** (a) TS and (b) BS of 24-h QPFs for day 1 (black), day 2 (red), and day 3 (blue) from the forecast made at
1200 UTC 7 Jun 2012 as a function of threshold (mm). (c),(d) to (m),(n) As in (a),(b), except for the forecasts made
at 1200 UTC of (c),(d) 8 and (e),(f) 9 Jun, at (g),(h) 0000 UTC and (i),(j) 1200 UTC of 10 Jun, and (k),(l) 0000 UTC
and (m),(n) 1200 UTC of 11 Jun, 2012, respectively. In left panels (for TS), the hit rate ($H/N$, %, rounded to integer)
at selected points and the classification group for each day are labeled. In right panels (for BS), the observed base
rate ($O/N$, %) and peak 24-h rainfall (mm) are also given.


In the four following forecasts made on 10-11 June (Figs. 7g-n), while the dependency on
event magnitude also exists, the QPFs made for A+ periods tend to have higher TSs above 75-
100 mm at the shorter ranges (Figs. 7g,i,k), as mentioned. All of good forecast quality (cf. Figs.
6a,b, columns 4-7), the TSs of these day-1 QPFs can be as high as 0.48 at 250 mm and 0.40 at

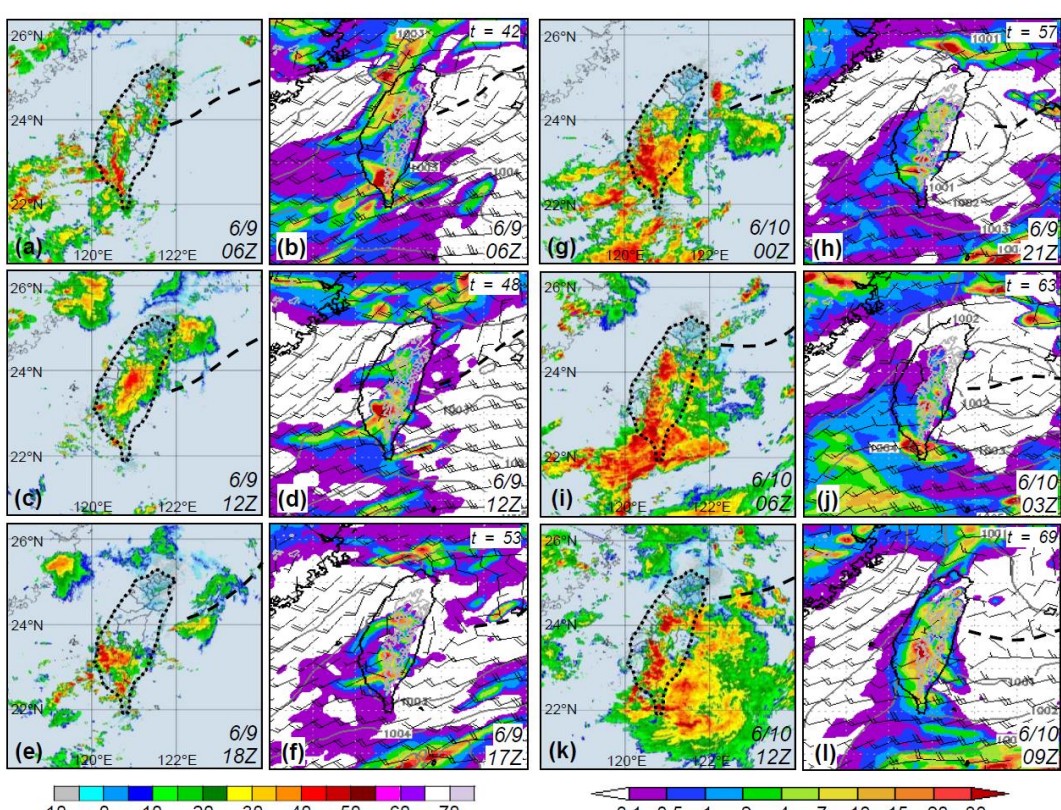

**Figure 8.** (First and third column) Radar reflectivity VMI composite (dBZ, scale at bottom left) in the Taiwan area
(width roughly 600 km) every 6 h from (a) 0600 UTC 9 Jun to (k) 1200 UTC 10 Jun, 2012 (original plots provided
by the CWB). (Second and fourth column) The CReSS forecast, starting from 1200 UTC 7 Jun 2012, of sea-level
pressure (hPa, every 1 hPa, over ocean only), surface wind (kts, barbs, at 10 m), terrain height at 1 and 2 km (gray
contours), and hourly rainfall (mm, color, scale at bottom right) valid at the time or within 3 h of the radar composite
as labeled [in UTC (forecast time in h) at lower (upper) right corner] over the same area. The thick dashed lines
mark the position of surface frontal or wind-shift line, based on NCEP gridded analyses for the observation (outline
of Taiwan also highlighted).




500 mm. At 350-500 mm, such high TS occurs with $O/N$ below < 10% (or even only 1%), and
thus indicates remarkable model accuracy in predicting the peak rainfall at the correct location in
the mountains in this event. Over thresholds ≥ 200 mm, BS values in Fig. 7 indicate that under-
prediction for this extreme event occurs much more often than over-prediction, while they also
tend to be closer to unity (with less under-forecast) for QPFs achieving higher TSs. An over-
prediction is more likely to happen for smaller events (A or below), across low thresholds below
50 mm, and/or when the rain area becomes small. In Fig. 7, for example, BS ≥ 2 at high
thresholds for A+ group occurs only when $O/N$ approaches zero (Figs. 7h,n), with the lone
exception in Fig. 7l on day 2. Overall, the model does not have a tendency to over-predict such a
large event (cf. Fig. 6).

The forecast on days 1-2 produced by the run starting at 1200 UTC 10 June is compared

with radar observations in Fig. 9. Together with Fig. 8, the radar panels cover the wettest 72 h
(0600 UTC 9-12 June) of the entire event. During day 1 (Figs. 9a-h), the scenario remains
similar to Fig. 8, and the model again was able to capture the mountain rainfall. The convection
moving in from the Taiwan Strait, however, was too active and the rain along the western coast
on day 1 was over-predicted with BSs ≈ 1.2-1.6 from 0.05 up to 100 mm (cf. Fig. 7j). Note that
in Fig. 7, some over-prediction across low thresholds can also exist for group A+ and lowers the
TS, which otherwise can often exceed 0.8 at and below 25 mm. In any case, the model's
performance over the low thresholds is of secondary importance.

Since 1200 UTC 11 June, the mei-yu front gradually approached northern Taiwan, and its

western section moved rapidly across the island after about 0000 UTC 12 June (Figs. 9i-p).
Studied by Wang et al. (2016b), the heavy rainfall in northern Taiwan (during 1400-2400 UTC)
was caused by quasi-linear convection that developed south of the front (Figs. 9i,k,m), along a
convergence zone between the low-level flow blocked and deflected by Taiwan's topography,
and unblocked flow further to the northwest (but still prefrontal) in the environment (also e.g., Li
and Chen, 1998; Yeh and Chen, 2002; Chen et al., 2005; Wang et al., 2005). In the model
forecast, with apparent errors in the position and moving speed of the front (Figs. 9i-p), it is
highly challenging to produce a similar system at the correct location and time even when the
overall scenario surrounding northern Taiwan are reasonably predicted. In the simulation of




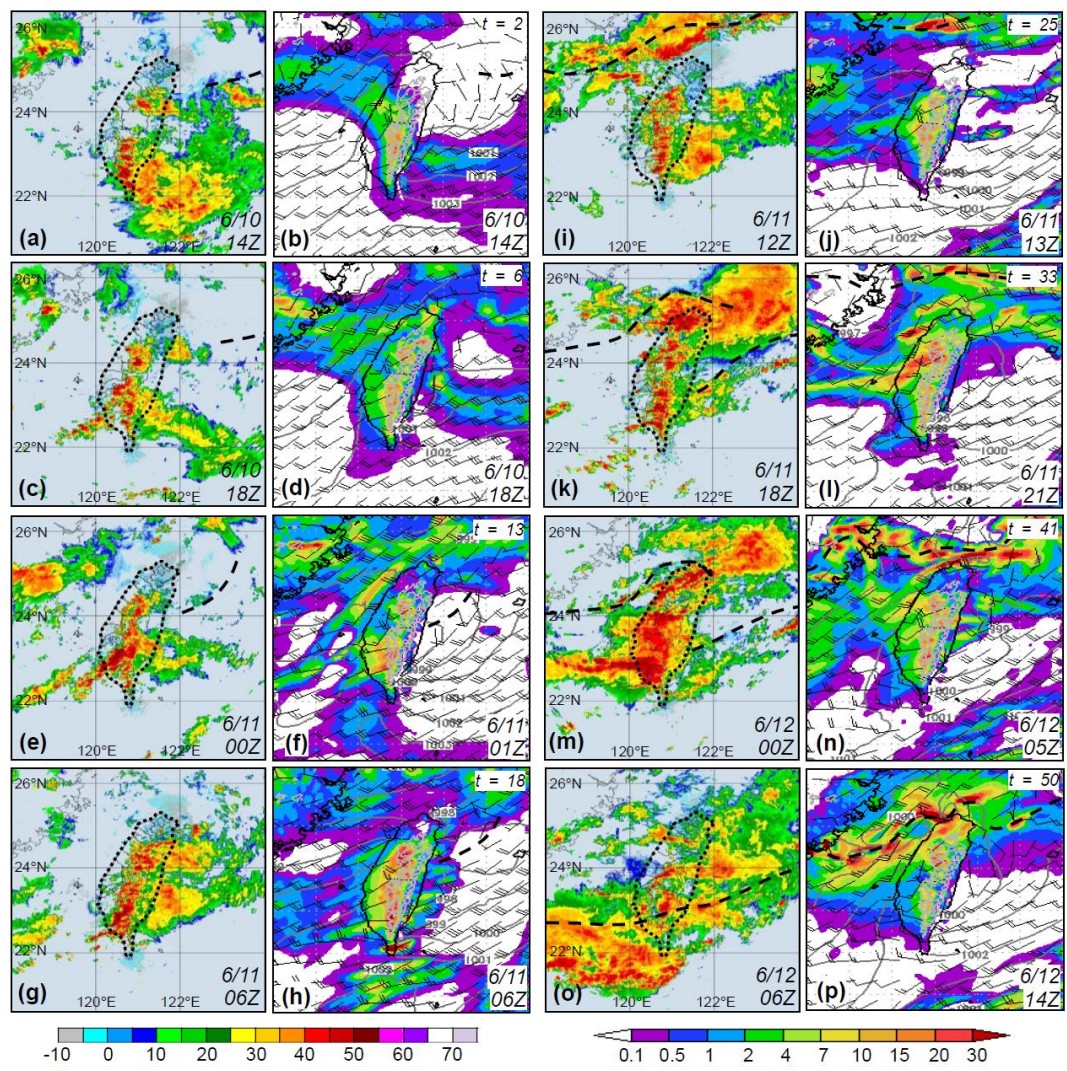

**Figure 9.** As in Fig. 8, but showing (columns 1 and 3) radar reflectivity composite (dBZ) at (a) 1400 UTC 10 Jun
and every 6 h from (c) 1800 UTC 10 Jun to (o) 0600 UTC 12 Jun, 2012, and (columns 2 and 4) the CReSS forecast,
starting from 1200 UTC 10 Jun 2012, of sea-level pressure (hPa), surface wind (kts), and hourly rainfall (mm) valid
at the time or within up to 8 h (towards the end) of the radar composite (as labeled).

Wang et al. (2016b), the rainbands cannot be fully captured even with a finer grid of $\Delta x$ = 1.5 km
and the NCEP final analyses as IC/BCs. Thus, although the model did indicate a real possibility
of heavy rainfall in northern Taiwan in Fig. 9, the high TS of 0.4 at 500 mm on day 2 (Fig. 7i)
came from the mountains, where the rainfall is clearly more predictable (cf. Figs. 6a,c, column




7), consistent with Walser and Schär (2004). Of course, the day-1 QPF with $t_0 = 1200$ UTC 11
June performed better in northern Taiwan than our example, but the goal here is to illustrate the
relatively high predictability of heavy rainfall phase-locked to the topography versus the low
predictability of rainfall produced by transient systems over low-lying plains.

The above example, together with other cases including those on 20 May 2013 and 20-21

May 2014, (cf. Table 3, not shown), suggests a lower predictability and a more challenging task
for QPFs produced by transient systems often in close association with the mei-yu front,
compared to topographic rainfall in Taiwan. Even though the overall scenario is reasonably and
realistically predicted (cf. Figs. 8 and 9), some position errors on the mei-yu front are almost
inevitable and the intrinsic predictability can limit the accuracy of the QPF. Also, for such
rainfall caused by transient systems, categorical statistics are known to be less effective in
verifying model QPFs (e.g., Davis et al., 2006; Wernli et al., 2008; Gilleland et al., 2010).
However, for the quasi-stationary, phase-locked rainfall over the topography in the majority of
large events (in both mei-yu and typhoon seasons, e.g., Chang et al., 1993; Cheung et al., 2008)
in Taiwan, they are still valid and useful as shown herein.

## 5 Dependency of Skill Scores to Event Size

In Section 3, a positive dependency in categorical measures by CReSS, including TS, POD,

and FAR, on rainfall amount is shown for the mei-yu regime in Taiwan, as predicted. Also
discussed in W15, this property arises mainly due to the positive correlation between the scores
and rain-area sizes, as given in Fig. 10 as an example with a correlation coefficient $r = 0.89$.
However, to explore whether the model is indeed more skillful in predicting larger rainfall
events, further analysis with the factor of rain-area size removed is needed. Different from W15,
our approach here is described below.

For each segment, the statistics ($H$, $M$, $FA$, and $CN$) at 13 fixed thresholds of 0.05-500 mm,

each occurring at a certain $O/N$ (if the threshold ≤ the observed peak amount), are known. The
observed base rate (0-100%) is divided into bins every 5% except at 0-5%, where it is subdivided
into 0-0.5, 0.5-2, and 2-5% to give more comparable sample size. For each group (A+ or A-D),

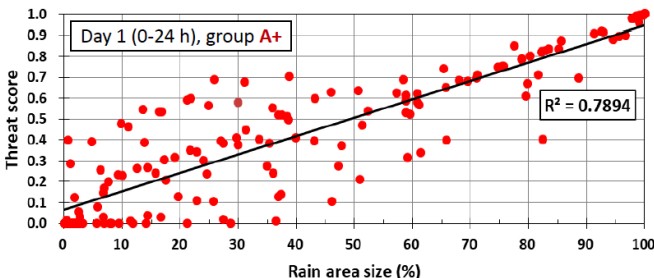

**Figure 10.** Scatter plot of TS versus observed rain-area size (%) from day-1 QPFs for group A+ from 0% to 100% (from high to low rainfall threshold). The square of correlation coefficient ($R^2$) is given.

the statistics are then summed for each bin regardless of their rainfall threshold. Thus, those in

the same bin come from rain areas with similar sizes. In Fig. 11a, the distribution of total counts

of thresholds across $O/N$ is plotted, and the larger events toward A+ are more capable to produce

rain areas larger in size (say, ≥60% of Taiwan). Also, the counts remain mostly around 50 for

$O/N \geq 40\%$, then rise to 200-300 with $O/N \leq 10\%$.

Due to fewer samples at larger $O/N$ values, the TSs for different groups (from a single $2 \times 2$

table for each bin) are presented only for $O/N \leq 40\%$ in Figs. 10b-d. While the scores for B-D are

roughly the same, the TSs for A are clearly higher compared to them on day 1, and those for A+

are again higher compared to A on days 1 and 2 over most part of this range, sometimes by 0.05-

0.1, when the factor of rain-area size is removed (Figs. 10b,c). On day 3 (Fig. 10d), however, the

TSs for larger events (A+ and A) show no particular advantage. Therefore, similar to typhoons in

W15, the 2.5-km CReSS is more skillful in predicting the larger mei-yu events in Taiwan within

2 days, over the heavy-rainfall area (again, more likely over the mountains).

The higher TSs and better skill for large events at $O/N$ within 40% (Fig. 10) are most likely

linked to the more favorable conditions at synoptic to meso-$\alpha$ scale, which the model is capable

to capture with higher accuracy (e.g., Walser and Schär, 2004). To briefly elaborate on this

aspect, seven items on the checklist used by CWB forecasters in the mei-yu season as a guidance

to issue heavy-rainfall warning (e.g., Wang et al., 2012a) are selected, and their occurrence

frequency, judged using surface weather maps and NCEP gridded analyses at the starting time of

each 24-h periods are summarized for different groups. These items include: 1) presence of

surface mei-yu front inside 20°-28°N, 118°-124°E; 2) Taipei (cf. Fig. 2) within 200 km south



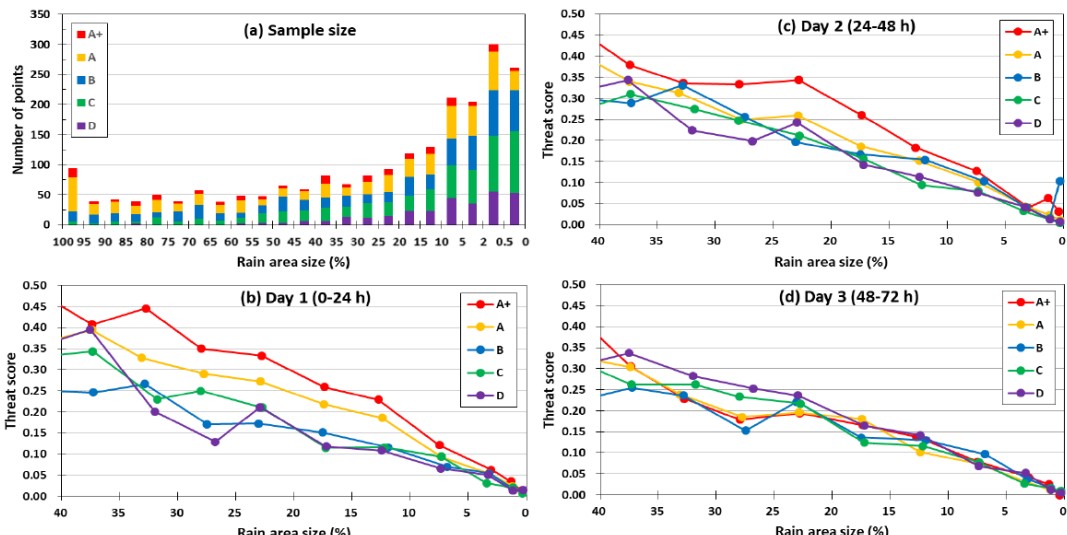

**Figure 11.** (a) The distribution of data points in the bins of observed rain-area size (%, every 5% from 100% to 5%, then 2-5%, 0.5-2%, and < 0.5%; same for days 1-3) among groups A+ and A to D. (b)-(d) The TS of 24-h QPFs for (b) day 1 to (d) day 3, respectively, as a function of observed rain-area size between 40% and 0%.

and 100 km north of the front; 3) Kaoshiung (cf. Fig. 2) within 200 km south of the front; 4) presence of low-level jet (LLJ) inside 18°-26°N, 115°-125°E at 850 or 700 hPa; 5) presence of mesolow near Taiwan; 6) Taiwan inside a low pressure zone; and 7) the mean sea-level pressure in Taiwan is below 1005 hPa. The results (Fig. 12) indicate that among the seven items, an average of 4.15 items are met in group A+, and this figure gradually declines toward smaller groups, from 3.67 in A, 2.84 in B, and finally to only 0.87 in X. Thus, as expected, the synoptic and meso-$\alpha$-scale conditions tend to be more favorable in larger events. Such conditions, in combination with orographic forcing in Taiwan, appear to favor good model performance, given a sufficient resolution.

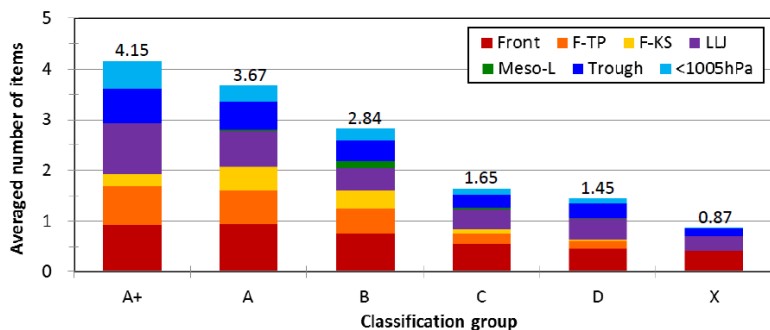

**Figure 12.** The average number in the seven items on the checklist met at the starting time of 24-h segments in
different classification groups, from A+, A to D, and X, with the proportion of each item plotted and the total
number labeled on top. Following the order (bottom to top), the seven items are: presence of surface mei-yu front
(front), front near Taipei (F-TP), front near Kaoshiung (F-KS), presence of LLJ, mesolow (meso-L), Taiwan inside a
low pressure zone (trough), and the mean sea-level pressure lower than 1005 hPa (<1005 hPa), respectively.

## 6 Summary and Concluding Remarks

In this section, our results are briefly summarized and the concluding remarks are given. In
the mei-yu seasons of 2012-2014, the overall TSs of day-1 QPFs for all events (no classification)
by the 2.5-km CReSS are 0.18, 0.15 and 0.09, respectively, at thresholds of 100, 250, and 500
mm. The corresponding TSs on day 2 are 0.13, 0.10 and 0.06, and 0.10, 0.03 and 0.00 on day 3.
Compared to previous results for mei-yu season in Taiwan (e.g., Hsu et al., 2014; Li and Hong,
2014; Su et al., 2016; Huang et al., 2016) and those from contemporary 5-km models, the results
herein show significant improvements by the 2.5-km CReSS, especially over the heavy-rainfall
thresholds at 130 mm and above. Moreover, when proper classification based on observed rain-
area size (i.e., event magnitude) are used, the CRM's ability to predict the extreme and top
events (group A+) are much higher, often by a factor of 2 or above in TS. For the top 4% and
most hazardous mei-yu events, our day-1 QPFs have TSs of 0.34, 0.24, and 0.16 at the same
thresholds of 100, 250, and 500 mm, respectively. The corresponding TSs are 0.32, 0.15 and
0.07 on day 2, and 0.25, 0.05 and 0.00 on day 3. Also, the QPFs for larger groups have higher
POD, lower FAR, and higher TS than smaller ones, across nearly all thresholds at all ranges of
days 1-3. Thus, the positive dependency in categorical scores on the overall rainfall amount also
exists in mei-yu regime in Taiwan, as predicted by W15 (and W16).



The improved performance by the 2.5-km CReSS in Taiwan, as shown by an example case,
lies in an improved ability to capture the phase-locked topographic rainfall at its correct location
in big events toward the extreme thresholds (350-500 mm). For mountain rainfall, it is more
predictable using a CRM that also better resolves the terrain (e.g., also Walser and Schär, 2004;
Roberts and Lean, 2007), and such QPFs with high hit rates are clearly very useful for hazard
mitigation. In contrast, the concentrated rainfall caused by transient systems (such as frontal
squall lines) has low predictability due to nonlinearity, and is notoriously difficult and highly
random to forecast at the correct location even though a realistic scenario is produced and the
potential of heavy rainfall indicated, unless at short-enough ranges. For such rainfall, model
QPFs are still informative and useful, but the categorical statistics may not be. As the high-
resolution models may possess a higher QPF skill in categorical statics for extreme events than
ordinary ones, at least for regions like Taiwan as demonstrated here (and in W15), it is also
recommended that such events should be examined with caution and proper classification.

**Code and data availability**

The model used in this study is called Cloud-Resloving Storm Simulator and its website for
downloading model and user's guide is at http://www.rain.hyarc.nagoya-
u.ac.jp/~tsuboki/cress_html/index_cress_eng.html. And the rainfall figures of Model is at the
website http://cressfcst.es.ntnu.edu.tw/.

**Author contribution**

Chung-Chieh Wang designed the experiments and Pi-Yu Chuang carried them out. Chih-
Sheng Chang operated the real-time model forecasting and Kazuhisa Tsuboki created the model
code. Shin-Yi Huang helped with some figures and Guo-Chen Leu provided some CWB results.
Chung-Chieh Wang prepared the manuscript with contributions from all co-authors.

**Competing interests**

The authors declare that they have no conflict of interest.



**Acknowledgments**

The first author, CCW, wishes to thanks assistants Ms. Y.-W. Wang, Mr. T.-C. Lin, and Mr. K.-Y. Chen for their help on this study. The CWB is acknowledged for providing the observational data, the radar plots, and the QPF verification results in Fig. 5. The National Center for High-performance Computing (NCHC) and the Taiwan Typhoon and Flood Research Institute (TTFRI) provided the computational resources. This study is jointly supported by the Ministry of Science and Technology of Taiwan under Grants MOST-103-2625-M-003-001-MY2, MOST-105-2111-M-003-003-MY3, MOST-108-2111-M- 003-005-MY2, and MOST-109-2625-M-003-001.

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
