# Peer review of "Evaluation of Mei-yu Heavy-Rainfall Quantitative Precipitation 1"

_Natural Hazards and Earth System Sciences, 2020_

## Author Comment (AC1)

NHESS-2020-397

Authors' Responses to Reviewer 1 (RC1, anonymous)

Date: 4 August 2021

Title: Evaluation of Mei-yu Heavy-Rainfall Quantitative Precipitation Forecasts in Taiwan by
      A Cloud-Resolving Model for Three Seasons of 2012–2014

Authors: C.-C. Wang, P.-Y. Chuang, C.-S. Chang, K. Tsuboki, S.-Y. Huang, and G.-C. Leu

**1. Overall comments:**

In this study, the authors evaluated the performance of the quantitative precipitation forecasts (QPFs) by a high-resolution CRM during the mei-yu seasons of Taiwan in 2012-2014, using categorical statistics. The results showed that the QPF skill is better for larger precipitation events, and improved compared to previous results. In addition, case analysis indicates that the strength of the high-resolution CRM lies in an improved ability to capture smaller scale processes for the phase-locked rainfall systems. These findings verify that the high-resolution CRM has good potential application in actual QPF during the mei-yu seasons of Taiwan. However, some major issues need to be clarified.

**Reply:**

The positive view and constructive comments from this reviewer (Reviewer 1) are deeply appreciated, and the paper has been revised according to the comments from all reviewers. In the revised manuscript (color-coded version), the changes made in response to Reviewer 1, Reviewer 2, Prof. G. T.-J. Chen (community comment), and by ourselves (mostly minor changes in English) are marked in red, blue, green, and orange, respectively. A point-by-point response to each of the comments from this reviewer are given below following their order. In each point, how and where the revision is made in the text is also specified.

**2. General comments:**

1)   It needs to give more explanation for the novelty of this study. As mentioned in the introduction, the purpose of this study is to clarify the dependency property in categorical scores of QPF and whether the skill of the high-resolution CRM is better than those in previous studies, although the studied object is changed from typhoons to mei-yu systems. This purpose has been basically fulfilled by W15 and W16. Therefore, they should not be considered as the novelty of this study, unless the study can prove the CReSS is sensitive to different weather

systems. However, from the conclusions, the higher-resolution CReSS primarily improved the forecast skill of the phased-locked topographic rainfall, as it better resolves the terrain and related small scale processes, which means the improvement of QPF caused by this CRM is not attributed to a better capture of the evolution of mei-yu front.

**Reply:** In the revision, more explanation is provided for the novelty of this study as suggested. In the introduction section, we better clarified that "the main purpose of this study is three-fold: 1) to assess the skill of the 2.5-km CReSS in predicting mei-yu rainfall at a higher resolution than before, especially for heavy to extreme rainfall events, 2) to clarify whether the dependency property in categorical scores also exists in the mei-yu regime in Taiwan? and 3) if the QPFs by CReSS prove to be improved, why or where its strength lies?" (L85-89). As demonstrated in this paper, the heavy-rainfall QPFs in the mei-yu season can be improved by using high-resolution models, and the underlying reason is also examined. We believe that these are good merits of the paper worthy of publication. While also exist in the typhoon regime, the dependency property in mei-yu regime in Taiwan has not been shown until this paper. In other places in the text, similar changes are also made to provide better context for the novelty of the study (L658-660), as suggested. We also agree with the reviewer's interpretation on why heavy rainfall QPFs are improved using a CRM, and these views are incorporated into the text in the revision (L412-414; L485), along the lines as suggested.

2)  Regard the QPF skill of the CReSS on different categories of rainfall events, this study shows a better QPF skill for larger rainfall events. However, this phenomenon may also happen for other high-resolution models, as a higher resolution permits the model to capture more small scale processes to improve convection development, and thus, more rainfall production. To a certain extent, this can be indicated by Figure 3 which shows that the QPF skill of the "All" category (the black lines) has a smaller success ratio (about false alarm) than those of large rainfall categories (A and A plus; the orange and red lines) for high rainfall thresholds (such as larger than 100 mm). It means that the high-resolution CReSS not only produces larger rainfall for large rainfall events, which leads to a higher TS scores, but also produces larger rainfall for small rainfall events, which leads to a smaller success ratio. Thus, this study needs to clarify more about the advantage of the CReSS model, apart from the resolution.

**Reply:** We agree with the reviewer on this point. In the revision, the review's opinion in the SR (in Fig. 4) is incorporated into the text along the lines as suggested (L239-241). Also,

the reviewer's interpretation on why heavy rainfall QPFs are improved using a CRM, which we agree, are incorporated into the text in the revision (L274-275), along the lines as suggested.

3) As discussed in section 4, the QPF error is also attributed to the forecast error on the evolution of mei-yu front. This error may lead to a worse QPF, as the location and timing of large rainfall can be completely incorrect. What is the cause of this error? Is it related to the boundary condition or the domain processes simulated by the CReSS? The answer of this issue can clarify the novelty of this study, as it is about the QPF associated with the mei-yu front.

**Reply:** In Wang et al. (2016b), the position error of the mei-yu front is presumably linked to the IC/BCs, even with a higher model resolution. This is clarified in the revision, along the lines as suggested (L396-397; L399; L401-402).

4) The comparative analysis or verification is based on a key indicator, the TS score. However, how large the value of TS score could be defined as skillful or a good skill? The study mentioned that when TS is larger than 0.15 it can be indicated "some predictive skill" (in line 228). Is there any objective definition or reference from operational prediction to support that?

**Reply:** As the TS is typically used to indicate predictive skill in a relative sense, we could not find a fixed value to define it as being skillful in the literature. We think, for example, that it would not be fair to say that TS = 0.20 is skillful but TS = 0.19 is not. However, based on experience of the operational sector (Dr. Leu, our last author, is the director of the Meteorological Forecast Center at the CWB) and some previous studies (e.g., Chien et al., 2002, 2006, as cited in text), a value like 0.15-0.2, which is above zero to a considerable degree, can be used to indicate some predictive skill as stated in the text. Throughout the text, it is revised to use TS more in a relative term, along the lines as suggested (L210-211).

**3. Specific comments:**

1) There are many places in the article that are not clearly expressed or improper use of vocabulary, which require major revisions. For examples (not exhausted), the sentences in lines 27-28 ("weaker events"->"smaller rainfall events"?), 35-40 ("where"->"when"?), 68 ("to hit"-> "that hit"?), 77 ("or event magnitude"->"or rainfall magnitude"?), 89 ("whether …" ?), 110-114 ("which are also run …"->

"which are applied for the model run …"?), 117-118 ("doubled the resolution"->"increase the resolution"?), 125-130 ("used include …"-> "used for QPF verification include …"?).

**Reply:** In the revision, all the above instances are corrected as suggested or modified to improve their clarity similar to suggested (L24; L36; L71; L79; L86-89; L119-120; L123; L126-128). Other modifications are also made throughout the text (in orange).

2) The manuscript uses too many abbreviations, which makes the readers hard to get the meaning of the sentences conveniently. Please delete the abbreviations which appear not frequently in the manuscript.

**Reply:** In the revision, the abbreviations not frequently used (such as NWP, W16, SST, VMI, … etc.) are deleted to improve the readability, as suggested.

3) Figure 3: Please explain more why after the rainfall events have been categorized into different rainfall magnitude events (A-D), different rainfall thresholds are still needed for each magnitude event.

**Reply:** In this study, wide range of rainfall thresholds (per 24 h) are chosen to fit rainfall events at different magnitudes. In the revision, this point is better clarified (L163; L184; L206-207), along the lines as suggested.

4) Figure 5: Why not put the CReSS results along with these model results for comparison? Are these models at a resolution of 5 km? If so, the comparison in the TS scores between the CReSS and these models is discounted, as their resolution are different.

**Reply:** We agree with the reviewer that the comparison in the TSs between the 2.5-km CReSS and the 5-km models is not very fair, as their resolution are different. In the revision, we therefore have moved Fig. 5 from Section 3.2 to the introduction section (and become Fig. 2) to be part of the review in research background (L91-96). This way, a direct comparison is avoided and we also stress the difference in model resolution when needed (L57-67), along the lines as suggested.

5) Lines 556-557: Did these previous results come from forecasts of an equal resolution (2.5 km)?

**Reply:** In the revision, it is better clarified that these results are from models at lower resolutions (L473), as suggested, and therefore the heavy-rainfall QPFs during the mei-yu season in Taiwan can be improved by using a higher resolution model like the 2.5-km CReSS. This is one of the main points of the study.

---

## Author Comment (AC2)

NHESS-2020-397

Authors' Responses to Reviewer 2 (RC2, anonymous)

Date: 2 August 2021

Title: Evaluation of Mei-yu Heavy-Rainfall Quantitative Precipitation Forecasts in Taiwan by
     A Cloud-Resolving Model for Three Seasons of 2012–2014
Authors: C.-C. Wang, P.-Y. Chuang, C.-S. Chang, K. Tsuboki, S.-Y. Huang, and G.-C. Leu

**1. Overall comments:**

This paper evaluates the performance of a convection-permitting model (the Cloud-Resolving Storm Simulator; CReSS) in simulating heavy precipitation over Taiwan during three mei-yu seasons in 2012, 2013 and 2014. The simulations are validated against rain gauges, radar data and NCEP analyses. A well-chosen classification criteria for the events is suggested, dividing the data into segments depending on whether at least 10% of the gauges showed heavy, moderate, some or no rain. For the validation, the authors employ several verification metrics based on contingency tables, namely Threat Score (TS), Probability of Detection (POD), False Alarm Ratio (FAR) and Bias Score (BS).

Although the methods and findings of the manuscript are not totally ground-breaking or unknown to the scientific community, the study presents a compelling analysis of Quantitative Precipitation Forecasts (QPFs) validation. Provided the evaluated model works at a convection permitting resolution and the investigation region is prone to suffering heavy precipitation events with high impact, the manuscript can be relevant to scientists working on the topic as well as stake-holders dealing with the impacts of such events. Therefore I believe there is interest in its publishing although it requires major revisions.

**Reply:**

The positive view and constructive comments from this reviewer (Reviewer 2) are deeply appreciated, and the paper has been revised according to the comments from all reviewers. In the revised manuscript (color-coded version), the changes made in response to Reviewer 1, Reviewer 2, Prof. G. T.-J. Chen (community comment), and by ourselves (mostly minor changes in English) are marked in red, blue, green, and orange, respectively. A point-by-point response to each of the comments from this reviewer are given below following their order. In each point, how and where the revision is made in the text is also specified.

**2. General comments:**

1) The conclusions in Section 5 (also in the Abstract) should be further elaborated. There is no need in these sections to write again the numbers of TS, FAR, etc. (e.g. lines 554 to 555 or 563 to 564). By doing so the main conclusions of the paper are hidden e.g. that QPFs are clearly improved by the use of convection permitting in heavy precipitation situations. Please, rewrite the conclusions and the Abstract to clearly point out the findings of the paper.

**Reply:** Thank you for this suggestion. In Section 6, it is revised to convey the findings of this study more clearly to the readers, as suggested (L469-491). The section is now partitioned into three paragraphs, each stating the findings linked to the specific aim and purpose. In the revision, the actual score values are also reduced to just twice, one for the day-1 overall TS (all events) and the other for the day-1 TS for A+ group, to illustrate the dependency property and the scores for extreme events, also along the lines as suggested (L472-473; L478). Similarly, the abstract is also revised (L15-16) as suggested.

2) It is not demonstrated that the CReSS model does a better job in orographic precipitation (phase locked) situations compared to transient systems as its stated in the conclusions (Lines 573 to 574). I agree that presumably, predictability is larger in those situations, due to 1) the fact that stationary systems have larger intrinsic predictability and 2) the better representation of the model orography. However the paper does not demonstrate this aspect. The concept "predictability" is lightly used and no quantification is provided (see for instance Hochman et al., 2021 where intrinsic predictability is quantified using dimension and persistence metrics for the systems). Instead only one case is shown (09-10 Jun) and two more cases are mentioned (L480) but no results are provided. From the case shown (09-10) the larger predictability of orographic precipitation is assumed by the fact that the location of the precipitating front (convective line) is not well represented but the TS scores are high (TS=0.4). This is not sufficient proof. It is advised that, given the type of information and analysis provided, the paper focuses on the "accuracy" of the simulation avoiding the analysis on "predictability".

Reference:
Hochman, A., Scher, S., Quinting, J., Pinto, J. G., and Messori, G.: A new view of heat wave dynamics and predictability over the eastern Mediterranean, Earth Syst. Dynam., 12, 133–149, https://doi.org/10.5194/esd-12-133-2021, 2021.

**Reply:** Thank you for this suggestion and we agree with it. As pointed out by this reviewer, the concept "predictability" is lightly used and no quantification is provided in this paper, it has been revised to focus on the "accuracy" of the simulation and avoid the analysis on "predictability" as suggested (L486-487). The reference of Hochman et al. (2021) is cited (L577-579), and some of the views of this reviewer are also incorporated into the text in the revision (L407-408). It is also better clarified in the revision that in our example case (Section 4), since the CReSS model produces most of the rainfall over 300 mm in the mountain regions (Fig. 6), the majority of the hits in this event must also occur in such regions at and above this threshold, along the lines as suggested (L327-329). Therefore, the evidence indicates that the hits occur mostly in the mountain regions, and this conclusion is not deduced just through the fact that the frontal location in our example is not well predicted.

**3. Specific comments:**

1) Title and Abstract (L18): It is highly possible that the reader is not familiar with what the Mei-Yu season is and when it occurs. Please include this information.

**Reply:** In the revision, it is added that the mei-yu season is May-June in the abstract, as suggested (L14).

2) L20-L21, L26-27, L71-73, L79-80: A perfect forecast would show a TS of 1. Why are TS values close to 0.1 a good result then? To support this statement either provide the information about the skill for that score or provide the TS values of the "past results and 5-km models". Since at this point of the paper the TS has not yet been defined please also include the information that a perfect forecast has a TS=1.

**Reply:** While the perfect value is 1, what constitutes a good TS value is mainly based on past results in the literature and experience of the operational sector (Dr. Leu, our last author, is the director of the Meteorological Forecast Center at the CWB). Some of these previous studies (e.g., Chien et al., 2002, 2006; Chien and Jou, 2004; Yang et al., 2004) are cited in the text to provide a proper context (L47-50). Also, it is revised to provide the range of TS ($0 \leq TS \leq 1$) at its first appearance as suggested (L173-174).

3) L88. The "dependency property" regarding the link between large events and improvement of the QPFs has not yet been explicitly explained. If I understood correctly these are defined in the papers W15, W16. A brief explanation of what

this is, is required here.

**Reply:** The dependency property is explained briefly at the beginning of this paragraph, as suggested (L78-85).

4) L89: The subobjective "further evaluate the model QPFs for larger and extreme events" should be part of the purpose 1).

**Reply:** Moved to purpose 1) in the revision, as suggested (L87).

5) L100-101 and L113: What do you mean by "CReSS needs no nesting". Dynamical downscales always need initial and boundary conditions from other, coarser, model".

**Reply:** In the revision, this sentence is reworded to "… a single domain without nesting" for better clarity, as suggested (L103).

6) L104-109: More information is needed about the parametrizations used. You need to explicitly mention the shallow convection parameterization scheme and the turbulence scheme. Also referring to Table 1 and W15 and W16.

**Reply:** As suggested, more information is added and references to Table 1 and W15 are also made in the revision (L106-124).

7) Table 1: How is the turbulence closure treated? Some models use a TKE 1D parametrization, others use a 3D, etc. What is the case in your simulations?

**Reply:** It is clarified that a 1.5-order closure is used in the PBL scheme with TKE prediction in the text and in Table 1, as suggested (L110-111; L114).

8) L228: Why is TS > 0.15 the threshold for predictive skill? Please explain, also if this information comes from previous literature provide the corresponding references.

**Reply:** In the revision, several past studies that use a similar TS value to indicate "some predictive skill" (e.g., Chien et al., 2002, 2006, Chien and Jou, 2004; Yang et al., 2004) are cited here, as suggested (L47-50).

9) L297-303: The statement "In model forecasts, when errors grow and the evolution deviates, the chance to become less rainy (not as favourable) is higher than more rainy, more so in forecasts made earlier at longer ranges." needs demonstration, either from literature or results.

**Reply:** We agree with this reviewer on this point. Thus, in the revision, this part of text is revised to "…This indicates that for larger events, the error growth with lead time in the model tends to become less rainy, as reflected in the decrease in BS…" to say only what is shown in the figure, as suggested (L259-261).

10) Figure 5: Could the authors elaborate on why are the scores lower for the events during June 2013 (either Day 1 or 2)? This aspect should also be included in the manuscript.

**Reply:** In the revision, several earlier studies are cited to provide likely reasons here, as suggested (L67-68).

11) Figure 7: In some panels, it seems as though the scores (TS or BS) are better on the third day of the forecast (day 3, blue line) than for the previous 2 days. Could you please explain this behaviour?

**Reply:** This shows essentially the dependency property, where the rainfall amount (event magnitude) apparently acts as a stronger influencing factor to the skill scores than other factors such as the forecast range. In the revision, this point is better clarified as suggested (L345-346).

12) Figure 12: The understanding of the Figure, its caption and explanation provided in the text is incomprehensible. Please rewrite. Why is the number of appearances of the different weather types, proof of the better model performance? Besides, large precipitation totals are usually linked to large-scale systems rather than localized convection, this is already known to the scientific community.

**Reply:** In the revision, the caption of Fig. 12 has been rewritten for better clarity as suggested (L463-467). The purpose of Fig. 12 is to link the event size (which has a positive relationship to QPF skill) to synoptic factors, as reflected by the average number of items met on the checklist used to facilitate heavy-rainfall forecasting at the CWB. Part of the relevant description is also revised to improve the readability as suggested (L453-457).

13) Summary and Concluding Remarks: Please recap the aim of the paper, main steps carried out and briefly the relevance of the study before enumerating the conclusions.

**Reply:** In the revision, the aim and main steps of the study are restated at the beginning of Section 6, as suggested (L469-471).

14) L556: Please, again, include what do you refer to by "compared to previous results".

**Reply:** In the revision, it is clarified that the "previous results" mean those reviewed in Section 1 (including Fig. 2), and several of those papers are also referenced here (L473-475), along the lines as suggested.

15) L564: When you mention "larger groups" are you referring to events with large coverage? Please reword.

**Reply:** Revised to "…the QPFs for larger events …" for better clarity, as suggested (L476-480).

16) L573: The statement "and such QPFs with high hit rates are clearly very useful for hazard mitigation." is not documented by your investigation. This was not shown in the paper. Delete.

**Reply:** Deleted as suggested (L485).

**4. Writing comments:**

1) L17: What does in real-time mean? Please consider deleting.

**Reply:** Deleted as suggested (L13).

2) L33: Should read: "Quantitative Precipitation Forecasting (QPF)…"

**Reply:** Changed as suggested (L32).

3) L37: Should read: "… mainly during two periods …"

**Reply:** Revised as suggested (L35).

4) L63-64: Delete "While the scores at the CWB will be compared with our results later," and "obviously much".

**Reply:** Deleted as suggested (L68-69).

5) L68-69: Model resolutions of 2.2 km are already being used in operational centres, for example de German Weather Service not only in research.

**Reply:** We agree with the reviewer. Here, the text is revised to "… more comparable to research, …", along the lines as suggested (L72).

6) L78: Change "more rain" by "the larger the rain".

**Reply:** Changed as suggested (L79).

7) L84: Wang (2015) has already been defined as W15. Please correct.

**Reply:** Revised similar to suggested (L84).

8) L85-86: Delete "For mei-yu rainfall in Taiwan, we are certainly keen to find out how this CRM performs, especially for the extreme events."

**Reply:** Deleted as suggested (L85).

9) L90-91: Delete "To answer these questions above are our objectives".

**Reply:** This sentence is revised according to the comment from Reviewer 1 (L86-89).

10) L113-114: Delete "which are also run four times a day, each out to 72 h (now 78 h)."

**Reply:** This sentence is revised according to the comment from Reviewer 1 (L119-120).

11) L116: Change: "highly dictated" by "forced".

**Reply:** Changed as suggested (L122).

12) Table 1: Provide the grid spacing of the topography in km as well.

**Reply:** Added as suggested (L114).

13) L135-136: Include this information about the relevance of the study in the abstract.

**Reply:** Included as suggested (L15-16).

14) L189.: Include the information that BS=1 implies no biases and that BS>(<)1 implies overestimation (underestimation) of the events.

**Reply:** Included as suggested (L175-176).

15) L218-236: Why is the explanation of Fig. 4 before the results concerning Figure 3?

**Reply:** Figure 3 has already been referenced both in the previous paragraph and in the earlier part of this paragraph (L192-201).

16) L276: Rephrase "…under-prediction for low…"

**Reply:** Revised as suggested (L245).

17) L295: The word "serious is not appropriate in this context"

**Reply:** The word "serious" is changed to "evident" along the lines as suggested (L257).

18) L447: Reword: "… June are compared…"

**Reply:** Revised to "The forecasts … June are compared…," as suggested (L380).

19) L496: Rephrase: "… sizes, as shown in Fig. 10…" and "… as an example…" by illustrated.

**Reply:** Revised to "… as illustrated in Fig. 10 … for the mei-yu regime" along the lines as

suggested (L423-424).

20) L580: The sentence "it is also recommended that such events should be examined with caution and proper classification. " does not bring any information and its colloquial.

**Reply:** The sentence is revised to "…, and can be helpful to hazard preparation and mitigation, along the lines as suggested (L491).

---

## Author Comment (AC3)

NHESS-2020-397

Authors' Responses to Prof. G. T.-J. Chen (Community Comment)

Date: 5 August 2021

Title: Evaluation of Mei-yu Heavy-Rainfall Quantitative Precipitation Forecasts in Taiwan by A Cloud-Resolving Model for Three Seasons of 2012–2014

Authors: C.-C. Wang, P.-Y. Chuang, C.-S. Chang, K. Tsuboki, S.-Y. Huang, and G.-C. Leu

**1. Overall comments:**

The purpose of this paper in to assess the skill of the 2.5 km CReSS in predicting mei-yu rainfall, to evaluate the model QPFs for larger and extreme events, as well as to understand the QPF strength of CReSS. The paper is well written and the results are of academic and application values. The paper can be accepted to be published in "Natural Hazards and Earth System Sciences". Some comments and suggestions are as follows:

**Reply:**

The positive view and constructive comments from Prof. G. T.-J. Chen (community comment) are deeply appreciated, and the paper has been revised according to the comments from all reviewers and the community. In the revised manuscript (color-coded version), the changes made in response to Reviewer 1, Reviewer 2, Prof. Chen, and by ourselves (mostly minor changes in English) are marked in red, blue, green, and orange, respectively. A point-by-point response to each of the comments from this reviewer are given below following their order. In each point, how and where the revision is made in the text is also specified.

**2. Specific comments:**

1.  In the Abstract, 2nd paragraph, "… the TSs are shown to be higher and the model more skillful in predicting larger events …". The plausible physical explanations are needed.

**Reply:** In the revision, the third paragraph is merged with the second one in the abstract to offer a plausible physical explanation for the improved skill in heavy-rainfall QPFs by the CRM with a higher resolution, along the lines as suggested (L25).

2.  In the Abstract, 3rd paragraph, "The strength of the model lies mainly in the topographic rainfall in Taiwan rather than migratory events that are less

predictable". The plausible physical explanations are needed.

**Reply:** In the revision, this sentence is reworded to "With the convection and terrain better resolved, the strength of the model is found to lie mainly in the topographic rainfall in Taiwan rather than …" to offer the physical explanations more clearly, as suggested (L24-26).

3. Section 3.1, 2nd paragraph, "…, the TSs are higher and the skill better for larger events than smaller ones". The plausible physical explanations are needed.

**Reply:** Here in Section 3.1, the phenomenon of higher TSs (better skill in model QPFs) for larger events is first demonstrated in Fig. 4, and is referred to as the positive dependency of model QPF skill on event magnitude (i.e., rainfall amount). The plausible physical explanations are explored and discussed in Section 4, and we note this in Section 3.2 clearly, along the lines as suggested (L280).

4. Section 3.1, 3rd paragraph, "…, the model is more capable to produce hits toward the rainfall maxima,…". The plausible physical explanations are needed.

**Reply:** The plausible physical explanations are explored and discussed in Section 4, and we note this in Section 3.2 clearly, along the lines as suggested (L280). Please also see our reply to point #3 above.

5. Section 3.1, 4th paragraph, "…the model also produces higher POD and SR for larger events compared to smaller ones…". The plausible physical explanations are needed.

**Reply:** The plausible physical explanations are explored and discussed in Section 4, and we note this in Section 3.2 clearly, along the lines as suggested (L280). Please see our reply to points #3 and #4 above.

6. Chapter 5, 3rd paragraph, "…the 2.5-km CReSS is more skillful in predicting the larger mei-yu events in Taiwan within 2 days,…" The plausible physical explanations are needed.

**Reply:** In this paper, a plausible physical explanation for the improved skill in heavy-rainfall QPFs by the CRM with a higher resolution is mainly investigated and discussed in Section 4 through examples. In the revision, various places in both Section 4 and later

sections are modified to make our findings in this regard more clearly to the readers, as suggested (L328-329; L401-402; L412; L484-485; L487).

---

## Author Response (AR2)

NHESS-2020-397

Authors' Responses to Reviewer 2 (RC2, anonymous)

Date: 10 Oct 2021

Title: Evaluation of Mei-yu Heavy-Rainfall Quantitative Precipitation Forecasts in Taiwan by
    A Cloud-Resolving Model for Three Seasons of 2012–2014

Authors: C.-C. Wang, P.-Y. Chuang, C.-S. Chang, K. Tsuboki, S.-Y. Huang, and G.-C. Leu

**Reply:**

The efforts and comments from this reviewer (Reviewer 2) are deeply appreciated, and the paper has been revised accordingly. In the revision (color-coded version), the changes made in response to Reviewer 2 and by ourselves (mostly minor changes in English) are marked in blue and orange, respectively. A point-by-point response to each of the comments from this reviewer are given below following their order. In each point, how and where the revision is made in the text is also specified.

**1. General comments:**

1) In the Abstract and Conclusions sections, again, the values of the TS are given with no comparison to the values of previous studies to clearly demonstrate the benefit of the CrESS simulation, e.g. L17, L472-L473. Although the TS values of previous studies are referenced in the introduction, the scores of TS should also be included in the Abstract and Conclusions to assert the real improvement of the CreSS simulations. Either write explicitly the values of TS from the previous experiment or provide the relative increase of TS in the CreSS simulations with respect to the previous papers.

**Reply:** Thank you for this suggestion. The TS values of previous studies are explicitly indicated (TSs < 0.1 at 100 mm and nearly zero at 250 mm and beyond) in both the Abstract and Section 6 (summary and conclusion) to indicate the improvement of the 2.5-km CReSS simulations, as suggested (L19, L461-462).

2) The concepts, score and skill of a model are confused in the manuscript. Scores (TS, POD, FAR) are a quantitative metric of the accuracy of a forecast. Skill of a model is the relative improvement of a QPF with respect to a reference value (most of the times a climatology). The metrics used for verification of the QPFs do not provide a measure of the skill of the model. All mentions to the "skill of the model" should be changed by mentions to the "accuracy of the model".

**Reply:** Thank you for this suggestion and we agree with it. In the revision, the word "skill" is replaced by "accuracy" (or "ability" or "performance" in some instances) to reference the categorical metrics such as the TS values (L53, L76, L86, L162, L191, L193, L198, L216, L229-231, L284, L334, L336, L409, L476), and retained only when the scores are compared to a reference to show improvement, as suggested.

3) In Fig. 12, an estimation of the number of items per weather type is introduced. While this graph clearly shows that large-scale systems (Meso-L, Through, Fronts) are associated with a larger number of items, this does not imply that "such conditions […] appear to allow for better model performance in QPFs". That particular aspect is not assessed. Either provide an assessment of the validation metrics (TS, POD, etc.) for each studied weather pattern (Fronts, Meso-L, Throughs etc.) or delete the figure 12 and lines L453 to L457, i.e., the complete sentence "The results (Fig. 12) … given a sufficient resolution".

**Reply:** Thank you for this suggestion. In the revision, the sentence is modified to say only what can be concluded from our analysis linked to Fig. 12, and is therefore revised to "… the synoptic and meso-$\alpha$-scale conditions tend to be more favorable in larger events, which in general also correspond to higher TS values (Figs. 3 and 11) in combination with the orographic forcing in Taiwan", along the lines as suggested (L444-445).

**2. Specific comments:**

1) L109 – Normally at the resolutions of your simulations (2.5 km) shallow convection needs to be parameterized because the model cannot resolve it explicitly. What do you mean by "without any cumulus parameterization scheme"? Is there any active shallow convection scheme? Please provide an explanation in the manuscript.

**Reply:** In the revision, it is clarified that no cumulus (or shallow convection) parameterization is used in the CReSS model, as suggested (L109-110).

2) L200 & L279 – In atmospheric sciences, one can only say that a change is significant after performing a statistical test, for example the t-test. Either provide assessment of the significance of your changes or rephrase these sentences.

**Reply:** In the revision, the word "significantly" is changed to other words such as "clearly" or

"considerably", along the lines as suggested (L200, L276).

3) L211 – Please provide the physical/mathematical explanation suggested by Chien et al., (2002, 2006), Chien and Jou (2004) and Yang et al., (2004) that a value of TS >=0.15 is a threshold for an accurate forecast and include that information in the manuscript. Also please adapt the word skill for accuracy (see general comment).

**Reply:** The value of TS $\geq$ 0.15 was used in some previous studies based on experience (without physical explanation), and we adopt the same criterion here. Because this standard may be somewhat arbitrary, the sentence is revised to "… if we select TS $\geq$ 0.15 to indicate some level of accuracy…", to soften the tone, following the suggestion (L211).

4) L232 – How come that at 130 mm a TS=0.07 implies a good forecast, if the threshold for a good forecast is TS>=0.15? Is it because the TS threshold varies depending on the analysed precipitation intensity? Also please adapt the word skill for accuracy (see general comment).

**Reply:** In the revision, this sentence is modified to make no reference to the accuracy of the forecast, as suggested (L231-232).

**3. Writting comments:**

L32 – Should read: "Quantitative Precipitation Forecasting (QPF) …"
**Reply:** Revised as suggested (L31).

L35 – Should read: "… rather frequently, mainly during …"
**Reply:** Revised as suggested (L34).

L36 – Delete "when"
**Reply:** In the revision, the long sentence is broken into shorter sentences, and revised to "…Chang et al., 2013). The landslides and …" for better readability, similar to as suggested (L35).

L39 – Delete: "model"
**Reply:** Deleted as suggested (L39).

L57-L67 – Sentence is too long. Split into several sentences

**Reply:** In the revision, the long sentences are broken into shorter sentences, as suggested (L56-61).

L64: Should read: "… mean (WEPS) for thresholds between 50-200 mm …"

**Reply:** Revised as suggested (L64).

L72 – Should read: "… more comparable to research studies"

**Reply:** Revised as suggested (L72).

L74 – Should read: "… forecasts showed …"

**Reply:** Revised as suggested (L74).

L76 – Should read: "… is remarkably higher for typhoon …"

**Reply:** Revised as suggested (L76-77).

L89 – Delete complete sentence: "As none of these above questions … our objectives"

**Reply:** The sentence is deleted as suggested (L89).

L103 – Change "without nesting" for "without intermediate nesting"

**Reply:** Changed as suggested (L103).

L206 – Delete: "and the second question in our objectives is answered"

**Reply:** Deleted as suggested (L206).

L208 – Delete: "for example"

**Reply:** Deleted as suggested (L208).

L220 – Change: "at times, or do not drop at all" for "for"

**Reply:** Changed as suggested (L220).

L262-L264 – Delete complete sentence: "Thus, as exemplified … but small (unimportant) events."

**Reply:** Deleted as suggested (L261).

Figure 4 – Delete "rounded to two decimal places"

**Reply:** Deleted as suggested (L267).

L279-L280 – Delete: "Thus, the first objective of this study is fulfilled"
**Reply:** Deleted as suggested (L276).

L285 – Delete: "shown above"
**Reply:** Deleted as suggested (L282).

L286 – Delete: "for further examination and discussion"
**Reply:** Deleted as suggested (L283).

L287 -Delete: "with the corresponding scores" and "in detail"
**Reply:** Both deleted as suggested (L284).

L288-L289 -Delete: "thereby to shed light on the source of skill seen in Fig. 4"
**Reply:** Deleted as suggested (L285).

L292 -Delete: "and the model's performance in predicting this event is thus highly relevant"
**Reply:** Deleted as suggested (L288).

L295 -Delete: "Reminiscent to the season average (cf Fig.3)"
**Reply:** Deleted as suggested (L290).

L313 -Delete: "in the same column"
**Reply:** Deleted as suggested (L308).

L314 – Change "lengthy" for "long-lasting"
**Reply:** Changed as suggested (L309).

L319 – Change "… somewhat too early …" for " … somewhat earlier …" and "with apparent" for "showing"
**Reply:** Both instances changed as suggested (L314).

L322 – Delete "however"
**Reply:** Deleted as suggested (L316).

L324 – Change "Compare" for "Compared"
**Reply:** Corrected as suggested (L319).

L328 – Change "observation" for "observations"

**Reply:** Changed as suggested (L322).

L328 – Delete "must"

**Reply:** Deleted as suggested (L323).

L329 – Delete "in this event".

**Reply:** Deleted as suggested (L323).

L334 – L336 – Delete complete sentence: "With such information … can also be verified here".

**Reply:** The sentence is deleted, as suggested (L328).

L337 – Change "commencement" for "beginning"

**Reply:** Changed as suggested (L328).

L346 – Start new paragraph at "In Fig. 8".

**Reply:** Revised as suggested (L338-344).

Figure 7 - Description is unintelligible. Please rewrite. Use points and split into shorter sentences.

**Reply:** Rewritten following the suggestion (L346-350).

L361-362– Change "…, while the dependency on event magnitude also exists, the QPFs… " for "…,the dependency on event magnitude exists, but the QPFs… "

**Reply:** Changed as suggested (L351).

L362 – L363 – Delete "as mentioned. All of good forecast quality (cf Figs 6ab, columns 4-7)"

**Reply:** Deleted as suggested (L352).

L370 – Change "lone" for "only"

**Reply:** Changed as suggested (L360).

L413-L414 – Delete complete sentence "Thus, through … to a certain degree."

**Reply:** Deleted as suggested (L403).

L471 – Delete: "In this section, … remarks are given".
**Reply:** Deleted as suggested (L459).

L476 – L477 – Rephrase the sentence as "The ability to represent the extreme and top events (group A+) in terms of the TS are much higher when a proper classification based on observed rain area size (i.e., event magnitude) is used."
**Reply:** Revised as suggested (L464-465).

L478 – Delete "at the same thresholds"
**Reply:** Deleted as suggested (L466).

L482 – Change "Through further analysis of example case, the improvement…" for "For a selected case study, the improvement…"
**Reply:** Changed as suggested (L470).

L484 – Delete "stationary"
**Reply:** Deleted as suggested (L471-472).

L485-L486 – Change "the concentrated rainfall" for "the accuracy of QPFs for concentrated rainfall".
**Reply:** Changed as suggested (L473-474).

L486 – Change "… squall lines) is highly random and more difficult to predict…" for "… squall lines) could not be demonstrated probably due to the difficulty to predict…"
**Reply:** Changed as suggested (L474-475).

L 487-L488 – Delete " and the potential of heavy rainfall indicated (unless at short ranges). For such rainfall model QPFs are still informative and useful, but the categorical scores may not be."
**Reply:** Deleted as suggested (L476).

L489 – Change "may possess" for "showed a".
**Reply:** Changed as suggested (L476).

L490 – Change "skill" for "accuracy" (see general comment).
**Reply:** Changed as suggested (L476).

L 491- Change "ordinary ones at least for regions like" for "coarser resolution models over the orographic region of Taiwan"

**Reply:** Changed as suggested (L477).

---

## Author Response (AR3)

NHESS-2020-397

Authors' Responses to the Editor and Reviewer 2 (RC2, anonymous)

Date: 16 Nov 2021

Title: Evaluation of Mei-yu Heavy-Rainfall Quantitative Precipitation Forecasts in Taiwan by
A Cloud-Resolving Model for Three Seasons of 2012–2014

Authors: C.-C. Wang, P.-Y. Chuang, C.-S. Chang, K. Tsuboki, S.-Y. Huang, and G.-C. Leu

**Reply:**

The additional comments from the editor (Dr. J. G. Pinto) and Reviewer 2 are appreciated, and the paper has been revised following these suggestions closely. In the revision (color-coded version), the minor modifications made in response to the Editor and Reviewer 2 are marked in green and blue, respectively. A point-by-point response to each of the comments are given below following their order. In each point, how and where the revision is made in the text is also specified.

**Comments from the Editor:**

The paper has now been re-revised by one of the original reviewers. Additionally to his/her comments in the review, I would like to point out you have yet to provide an explanation of where the 0.15 threshold for TS come from or whether it has a mathematical or physical basis. This was asked previously by the reviewers but no further information was provided, just the references to the previous publications. Therefore, the paper is returned for a minor revision (by the editor)

**Reply:** The value of TS $\geq 0.15$ was used in some previous studies and based on experience (mainly in the operational sector), and we note in the revision that the value is perhaps somewhat arbitrary (L211), along the lines as suggested. The readers would understand that the model QPFs have the skill of TS $\geq 0.15$ at the specified thresholds and ranges stated in this sentence, and we do not imply anything beyond that.

**Comments from Reviewer 2:**

**Please include the following corrections:**

L19, L461: Please provide the exact value of TS for the 5 km simulations and the 250 mm threshold. An approximation in the form TS~0 "approach to zero" here does not make sense. Please provide exact result with two decimals precision, just as for the other values.

**Reply:** Thank you for this suggestion. In the revision, the TS values of previous studies are explicitly indicated as TS ≤ 0.02 at 250 mm (and beyond) in both places, as suggested (L19, L461).

L463. Change the word significant. It is not based on significance test analysis

**Reply:** In the revision, the word "significant" is changed to "considerable" as suggested (L463).